# Optimization-Based Defender via Coarse-To-Fine Tensor Network Representation

## Abstract

Deep neural networks are vulnerable to well-designed adversarial attacks. Although numerous defense strategies have been proposed, most are tailored to specific threats or datasets and thus struggle to generalize across diverse adversarial scenarios. In this paper, we propose Tensor Network Purification (TNP), a novel optimization-based defense technique built upon a specially designed tensor network decomposition algorithm. TNP depends neither on the pre-trained generative model nor the specific dataset, enabling robust generalization. To this end, the key challenge lies in relaxing Gaussian-noise assumptions of classical decompositions and accommodating the unknown perturbation distributions. Instead of imposing consistency by traditional objectives, TNP aims to reconstruct the latent clean example from its adversarially perturbed input. Specifically, TNP leverages progressive downsampling with a new adversarial objective that minimizes reconstruction error while suppressing the inadvertent restoration of the perturbations. Extensive experiments on CIFAR-10, CIFAR-100, and ImageNet show that TNP generalizes effectively across diverse norm threats, attack types, and datasets, delivering a versatile and promising defense.

## 1 Introduction

Deep neural networks (DNNs) have achieved remarkable success across a wide range of tasks. However, DNNs have been shown to be vulnerable to adversarial examples (Szegedy et al., 2014; Goodfellow et al., 2015), which are generated by adding small, human-imperceptible perturbations to natural images but completely mislead the prediction results of DNNs, potentially leading to disastrous consequences. This inherent vulnerability of DNNs underscores the critical need for robust defense mechanisms to effectively mitigate adversarial attacks.

In response, numerous methods have been proposed to defend against adversarial examples. Notably, adversarial training (AT, Goodfellow et al., 2015) typically aims to retrain DNNs using specific adversarial examples, achieving robustness to seen types of adversarial attacks but performing poorly against unseen perturbations (Laidlaw et al., 2021). Another class of defense methods is adversarial purification (AP, Yoon et al., 2021; Pei et al., 2025), which leverages pre-trained generative models to remove adversarial perturbations and demonstrates better generalization than AT against unseen attacks (Nie et al., 2022; Lin et al., 2024). However, AP methods heavily rely on pre-trained models tailored to specific datasets, limiting their transferability to different data distributions and tasks. As a result, both mainstream techniques face generalization challenges: AT struggles with diverse attacks, and AP with task generalization, restricting their deployment to broader adversarial scenarios.

To address these challenges, we aim to develop a novel adversarial purification framework built upon tensor network (TN) decomposition. As an optimization-based technique, TN depends neither on any pre-trained generative model nor specific dataset (Oseledets, 2011; Zhao et al., 2016), enabling it to achieve strong generalization across diverse adversarial scenarios. Serving as a plug-and-play pre-processing step, TN-based AP can eliminate potential adversarial perturbations from both clean and adversarial examples before they are fed into the classifier (Yoon et al., 2021). This implies that robust defense can be achieved without retraining the classifier model. Specifically, an input example is first transformed into a higher-order tensor representation, which is then decomposed into a set of low-rank tensor cores through TN decomposition. By optimizing these tensor cores under a specially designed constraint objective, the TN decomposition effectively suppresses the

undesirable perturbations while maintaining the essential semantic content of the image. Finally, the purified image is reconstructed from the optimized tensor cores and passed to the standard classifier for prediction. Moreover, since TN operates directly on each individual input without training of model parameters, it exhibits inherent resistance to adaptive adversarial attacks, as further discussed in Appendix C. Collectively, these advantages highlight TN-based AP as a highly promising direction, offering the transferability to be effectively applied across a wide range of adversarial scenarios.

However, classical TN methods are primarily designed for image completion and denoising tasks in which the corruption is sparse or follows a Gaussian distribution. In contrast, the distribution of adversarial perturbations fundamentally differs from these assumptions and often aligns with the intrinsic statistics of the data (Ilyas et al., 2019; Allen-Zhu & Li, 2022). As a result, these perturbations behave more like genuine features than noise, making them difficult to model explicitly and prone to being inadvertently reconstructed. To address this issue, we first explore the distribution changes of perturbations during the optimization process and initially mitigate their impact through progressive downsampling. Building upon these insights, we develop a coarse-to-fine TN incremental learning algorithm and introduce a novel adversarial optimization objective designed to avoid overly constraining the reconstruction within the perturbations. Through this pipeline, we propose Tensor Network Purification (TNP), which bridges the gap between classical low-rank TN representation with Gaussian noise assumption and the removal of adversarial perturbations with unknown distributions. Unlike classical TN methods applied to adversarial examples, TNP prevents naive low-rank representation of the perturbed input and encourages the reconstructed examples to approximate the unobserved clean examples.

We empirically evaluate the performance of TNP by comparing it with AT and AP across a wide range of attack settings using multiple classifiers on CIFAR-10, CIFAR-100, and ImageNet. TNP consistently demonstrates strong generalization across diverse adversarial scenarios. Specifically, it achieves a 26.45% improvement in average robust accuracy over AT across different norm threats, 9.39% over AP across multiple attacks, and 6.47% over AP across different datasets. Furthermore, compared to existing TNs, TNP more effectively removes adversarial perturbations while preserving consistency between reconstructions from clean and adversarial examples. These results collectively underscore the effectiveness and potential of TNP. In summary, our contributions are as follows.

- We propose an optimization-based technique built upon tensor network representation, which requires neither a generative model nor any training cost, enabling general-purpose adversarial purification.

- Based on our analysis of distribution changes in adversarial perturbations during optimization, we design a novel optimization objective for coarse-to-fine TN representation learning that prevents the restoration of adversarial perturbations.

- We conduct extensive experiments on various datasets, demonstrating that TNP achieves robust performance and exhibits strong generalization across diverse adversarial scenarios.

## 2 RELATED WORKS

**Adversarial robustness**  To defend against adversarial attacks, researchers have developed various techniques aimed at enhancing the robustness of DNNs. Goodfellow et al. (2015) propose AT technique to defend against adversarial attacks by retraining classifiers with adversarial examples (Wang et al., 2019; Tack et al., 2022). In contrast, AP methods (Shi et al., 2021; Srinivasan et al., 2021) aim to purify adversarial examples before classification without retraining the classifier. Currently, the most common AP methods (Nie et al., 2022; Bai et al., 2024) rely on pre-trained generative models as purifiers, which are trained on specific datasets and are hard to generalize to data distributions outside their training domain. Lin et al. (2024) propose applying AT (Zhang et al., 2019) technique to AP, optimizing the purifier to adapt to new data distributions, at the cost of substantial training costs. Although TNP employs AP technique, it fundamentally differs from these works in that an optimization-based framework relying solely on the information of the single input example for AP, without requiring any additional priors from pre-trained models and training costs.

**Tensor network and TN-based defense methods**  Tensor network (TN) is a classical tool in signal processing, with many successful applications in image completion and denoising (Kolda & Bader, 2009; Cichocki et al., 2015). Compared to classical TN methods such as TT (Oseledets, 2011)

and TR (Zhao et al., 2016), we employ the quantized technique (Khoromskij, 2011) and develop a coarse-to-fine strategy. Recent work (PuTT, Loeschcke et al., 2024) also employs a coarse-to-fine strategy, aiming to achieve better initialization for faster and more efficient TT decomposition by minimizing the reconstruction error. In comparison, our method progresses from low to high resolution, explicitly targeting perturbation removal and analyzing the impact of downsampling on perturbations. Furthermore, we propose a novel optimization objective that goes beyond simply minimizing the reconstruction error, focusing instead on preventing the restoration of perturbations.

With the growing concern over adversarial robustness, a line of work has attempted to leverage TNs as robust denoisers to defend against adversarial attacks. In particular, Yang et al. (2019) reconstruct images and retrain classifiers to adapt to the new reconstructed distribution. Entezari & Papalexakis (2022) analyze classical TNs and show their effectiveness in removing high-frequency perturbations. Additionally, Bhattarai et al. (2023) extend the application of TNs beyond data to include classifiers, a concept similar to the approaches of Phan et al. (2023); Rudkiewicz et al. (2024). Furthermore, Song et al. (2024) employ training-free techniques while incorporating ground truth information to defend against adversarial attacks. However, the aforementioned methods rely on additional prior knowledge or are limited to specific attacks. In this paper, we aim to achieve robustness solely by optimizing TNs themselves, establishing them as a plug-and-play and promising adversarial purification technique.

# 3 BACKGROUND

**Notations** Throughout the paper, we denote scalars, vectors, matrices, and tensors as lowercase letters, bold lowercase letters, bold capital letters, and calligraphic bold capital letters, e.g., $x$, $\boldsymbol{x}$, $\boldsymbol{X}$ and $\boldsymbol{\mathcal{X}}$, respectively. A $D$-order tensor is an $D$-dimensional array, e.g., a vector is a 1st-order tensor and a matrix is a 2nd-order tensor. For a $D$-order tensor $\boldsymbol{\mathcal{X}} \in \mathbb{R}^{I_1 \times \cdots \times I_D}$, we denote its $(i_1, \ldots, i_D)$-th entry as $x_{\mathbf{i}}$, where $\mathbf{i} = (i_1, \ldots, i_D)$. Following the conventions in deep learning, we treat images as vectors, e.g., input example $\boldsymbol{x}_{in}$, clean example $\boldsymbol{x}_{cln}$, adversarial example $\boldsymbol{x}_{adv}$, and reconstructed example $\boldsymbol{y}$.

**Tensor network decomposition** Given a $D$-order tensor $\boldsymbol{\mathcal{X}} \in \mathbb{R}^{I_1 \times \cdots \times I_D}$, tensor network decomposition factorizes $\boldsymbol{\mathcal{X}}$ into $D$ smaller latent components by using some predefined tensor contraction rules. Among tensor network decompositions, tensor train (TT) decomposition (Oseledets, 2011) enjoys both quasi-optimal approximation as well as the high compression rate of large and complex data tensors. In particular, a $D$-order tensor $\boldsymbol{\mathcal{X}} \in \mathbb{R}^{I_1 \times \cdots \times I_D}$ has the TT format as $x_{\mathbf{i}} = \boldsymbol{A}_{i_1}^1 \boldsymbol{A}_{i_2}^2 \ldots \boldsymbol{A}_{i_D}^D$, where $\boldsymbol{A}_{i_d}^d \in \mathbb{R}^{r_{d-1} \times r_d}$, for $d \in [D]$ and $i_d \in [I_d]$. Then, $(1, r_1, \ldots, r_{d-1}, 1)$ is the TT rank of $\boldsymbol{\mathcal{X}}$. For simplicity, we denote $\boldsymbol{\mathcal{X}} = \mathrm{TT}(\boldsymbol{\mathcal{A}}^1, \ldots, \boldsymbol{\mathcal{A}}^D)$. When each dimension $I_d$ of $\boldsymbol{\mathcal{X}}$ is large, quantized tensor train (QTT, Khoromskij, 2011) becomes highly efficient, which splits each dimension into powers of two. For example, a $2^D \times 2^D$ image can be rearranged into a more expressive and balanced $D$-order tensor. For brevity, hereafter, a $2^D \times 2^D$ image $\boldsymbol{x}_D$ shall be called a resolution $D$ image, whose quantized tensor is $\boldsymbol{\mathcal{X}}_D = \mathrm{Q}(\boldsymbol{x}_D)$. QTT core denotes the core tensor after decomposition.

# 4 METHOD

Tensor network (TN) is a classical tool in signal processing, with many successful applications in image completion and denoising. By leveraging the $\ell_2$-norm as the primary optimization criterion, which aligns well with the statistical properties of a normal distribution, these methods (Phan et al., 2020; Loeschcke et al., 2024) have demonstrated strong capabilities in removing Gaussian noise.

However, the distribution of well-designed adversarial perturbations is essentially different from Gaussian noise and cannot be modeled explicitly (Ilyas et al., 2019; Allen-Zhu & Li, 2022), which challenges the conventional assumptions of TN-based denoising methods, leading to ineffectiveness on adversarial purification for $\boldsymbol{x}_{adv}$. To minimize the loss $\|\boldsymbol{x}_{adv} - \mathrm{TN}(\boldsymbol{x}_{adv})\|_2$, TN decompositions fit all feature components of $\boldsymbol{x}_{adv}$, including the adversarial perturbations. However, in the presence of adversarial attacks, we aim to restore unobserved $\boldsymbol{x}_{cln}$ from the input $\boldsymbol{x}_{adv}$, that is: $\mathrm{TN}(\boldsymbol{x}_{adv}) \approx \boldsymbol{x}_{cln}$ rather than $\boldsymbol{x}_{adv}$. Based on the above analysis, it is crucial to overcome two challenges in designing an effective TN method: *Q1. How can we transform the non-specific adversarial perturbations to align with the modeling assumptions of TNs? Q2. How can we formulate an optimization process for TNs that avoids inadvertently restoring those perturbations?*

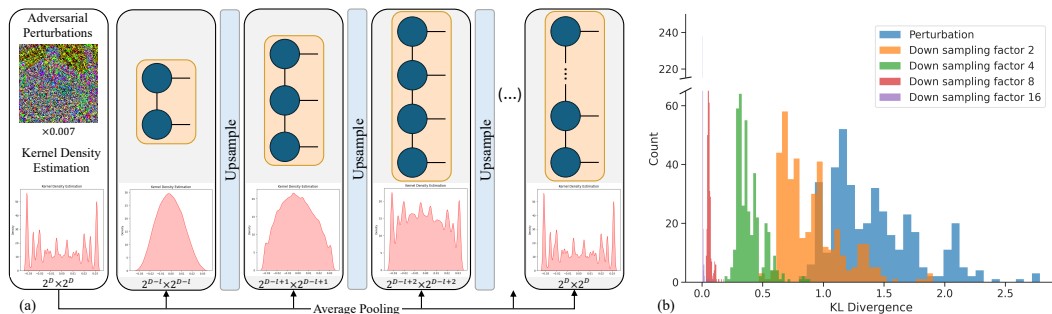

Figure 1: Compare the adversarial perturbations in the downsampled images. (a) The distribution changes of adversarial perturbations during downsampling process. (b) The KL divergence between the adversarial perturbations and the Gaussian distributions with the same sample mean and variance.

For *Q1*, we explore how adversarial perturbations behave under downsampling with average pooling in Section 4.1. Intuitively, the central limit theorem suggests that as an image is progressively downsampled, aggregated perturbations begin to resemble a normal distribution. Thus, even an $\ell_2$-based penalty becomes effective in suppressing the perturbations at coarse resolution.

However, while this insight helps suppress perturbations at lower resolutions, there remains the challenge of reconstructing the original resolution image. When upsampling and further optimizing using $\|\boldsymbol{x}_{adv} - \text{TN}(\boldsymbol{x}_{adv})\|_2$, the perturbations will still be restored. This connects with *Q2*, for which we design a new adversarial optimization objective, as shown in Section 4.3.

## 4.1 DOWNSAMPLING USING AVERAGE POOLING

An intuitive explanation for why downsampling aids in perturbation removal can be derived from the Central Limit Theorem (CLT, Grzenda & Zieba, 2008). When an image is downsampled by average pooling, the random components (e.g., pixel-level noise or minor adversarial perturbations) within those pooling patches are aggregated. We hypothesize that, given an adversarial example $\boldsymbol{x}_{adv}$, downsampling the $\boldsymbol{x}_{adv}$ from its original resolution $D$ to a lower resolution $D-1$ will smooth out the perturbations. As the downsampling process progresses further, the distribution of the aggregated perturbations in the coarse resolution image $\boldsymbol{x}_{D-l}$ is expected to converge toward a normal distribution, as illustrated in Figure 1a. More results are shown in Appendix F.

To investigate this hypothesis in real datasets, we measure the KL divergence between the histograms of adversarial perturbations and the Gaussian distributions with the same sample mean and variance across 512 images from ImageNet. As shown in Figure 1b, the distribution of those perturbations progressively aligns with that of Gaussian noise as the downsampling process progresses. Consequently, even a classical tensor network (CTN) can effectively remove or mitigate adversarial perturbations at coarse resolution. Additionally, we further compare the influence of different downsampling methods to underscore the advantages of average pooling, as discussed in Appendix A.

## 4.2 TENSOR NETWORK PURIFICATION

Building upon our downsampling-based intuition, we design a coarse-to-fine purification pipeline by extending PuTT (Loeschcke et al., 2024), which employs progressive downsampling for better initialization of QTT cores. The workflow of tensor network purification (TNP) for classification tasks is illustrated in Figure 2, where the quantized $\boldsymbol{\mathcal{X}} = \text{Q}(\boldsymbol{x})$, TT decomposition $\boldsymbol{\mathcal{X}} \approx \boldsymbol{\mathcal{Y}} = \text{TT}(\boldsymbol{\mathcal{A}}^1, \ldots, \boldsymbol{\mathcal{A}}^D)$, and reconstruction $\boldsymbol{y} = \text{Q}^{-1}(\boldsymbol{\mathcal{Y}})$ processes are depicted.

Initially, the $2^D \times 2^D$ input example $\boldsymbol{x}_D$ (potentially adversarial example $\boldsymbol{x}_{adv}$ or clean example $\boldsymbol{x}_{cln}$), whose quantized version is a $D$-order tensor $\boldsymbol{\mathcal{X}}_D$, is first downsampled to a resolution $D-l$ example $\boldsymbol{x}_{D-l}$, corresponding to a $(D-l)$-order tensor $\boldsymbol{\mathcal{X}}_{D-l}$. The QTT cores of $\boldsymbol{\mathcal{X}}_{D-l}$ are optimized by CTN via backpropagation within a standard reconstruction error $\|\boldsymbol{x}_{D-l} - \boldsymbol{y}_{D-l}\|_2$. Once the approximation of $\boldsymbol{\mathcal{X}}_{D-l}$ is stabilized, the prolongation operator $\boldsymbol{\mathcal{P}}_{D-l+1}$ is applied to the QTT format of $\boldsymbol{\mathcal{X}}_{D-l}$, producing a $(D-l+1)$-order tensor $\boldsymbol{\mathcal{P}}_{D-l+1}\boldsymbol{\mathcal{X}}_{D-l}$. Additionally, we define the linear function $\text{P}_d(\cdot)$ that acts at the image level, with the effect of upsampling from resolution $d-1$ to $d$,

---

**Algorithm 1** Adversarial optimization process.

**Input:** Example $\boldsymbol{x}_d$, number of iterations $T$,
steps $N$, scale $\alpha$ and $\eta$, learning rate $\beta$
Initialize $\boldsymbol{y}_d \leftarrow \mathrm{P}_d(\boldsymbol{y}_{d-1}), \boldsymbol{\delta}_d \leftarrow \boldsymbol{0}$
**for** $t = 1, 2, \ldots, T$ **do**
   **for** $n = 1, 2, \ldots, N$ **do**
      $\ell \leftarrow \mathcal{L}_{adv}(\boldsymbol{y}_d + \boldsymbol{\delta}_d, \boldsymbol{x}_d)$
      $\boldsymbol{\delta}_d \leftarrow \mathrm{clip}(\boldsymbol{\delta}_d + \alpha \mathrm{sign}(\nabla_{\boldsymbol{y}_d}\ell), -\eta, \eta)$
      $\boldsymbol{\delta}_d^* \leftarrow \mathrm{clip}(\boldsymbol{y}_d + \boldsymbol{\delta}_d, 0, 1) - \boldsymbol{y}_d$
      Gradient descent based on Eq. (1):
      $\boldsymbol{y}_d \leftarrow \boldsymbol{y}_d - \beta \nabla_{\boldsymbol{y}_d}\mathcal{L}_{tnp}(\boldsymbol{x}_d, \boldsymbol{y}_d, \boldsymbol{\delta}_d^*)$
**return** $\boldsymbol{y}_d$

---

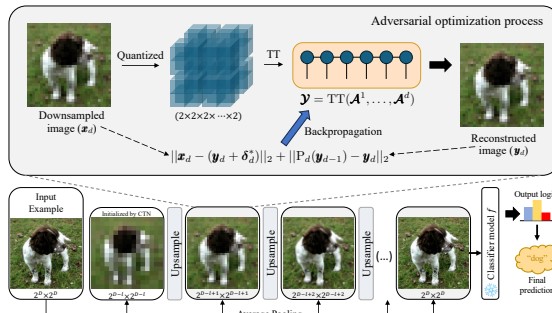

Figure 2: Illustration of tensor network purification.

details in Appendix B.2. This serves as an initialization to find the optimal QTT cores of $\boldsymbol{\mathcal{X}}_{D-l+1}$ and the reconstructed downsampled example $\boldsymbol{y}_{D-l}$.

Next, the input example $\boldsymbol{x}_D$ is once again downsampled to a resolution $D - l + 1$ example $\boldsymbol{x}_{D-l+1}$. At this stage, the QTT cores of $\boldsymbol{\mathcal{X}}_{D-l+1}$ are optimized using the adversarial optimization objective within a novel loss function as shown in Eq. (1). Similarly, once the approximation of $\boldsymbol{\mathcal{X}}_{D-l+1}$ stabilizes, the upsampling operation is performed. This process is repeated iteratively until reaching the QTT approximation $\boldsymbol{\mathcal{Y}}_D$ of the original resolution $\boldsymbol{\mathcal{X}}_D$.

Finally, TNP can purify potential adversarial examples ($\boldsymbol{x}_{cln}$ or $\boldsymbol{x}_{adv}$) before feeding them into classifier $f$, e.g., $f(\mathrm{TNP}(\boldsymbol{x}_{cln})) = f(\mathrm{TNP}(\boldsymbol{x}_{adv})) = gt$, where $gt$ is the ground truth label. As a plug-and-play module, TNP requires no modification to $f$ and can be integrated with any classifier.

### 4.3 ADVERSARIAL OPTIMIZATION PROCESS

Unlike traditional reconstruction, in the context of adversarial attacks, we can only observe the adversarial example $\boldsymbol{x}_{adv}$, while the goal is to reconstruct a clean $\boldsymbol{y}$ close to the unobserved clean example $\boldsymbol{x}_{cln}$. To bridge the gap between $\boldsymbol{x}_{adv}$ and $\boldsymbol{x}_{cln}$, we propose a new adversarial optimization objective that introduces an auxiliary variable $\boldsymbol{\delta}$. Moreover, we leverage the previously reconstructed downsampled example as a crucial "prior" to guide the approximation toward $\boldsymbol{x}_{cln}$.

Here, we outline the optimization procedure for $\boldsymbol{x}_d$, which corresponds to the gray box in Figure 2. Formally, given the resolution $d$ example $\boldsymbol{x}_d$, we attempt to obtain the reconstructed example $\boldsymbol{y}_d$ by performing gradient descent on optimization loss functions of

$$\mathcal{L}_{tnp}(\boldsymbol{x}_d, \boldsymbol{y}_d, \boldsymbol{\delta}_d^*) = ||\boldsymbol{x}_d - (\boldsymbol{y}_d + \boldsymbol{\delta}_d^*)||_2 + ||\mathrm{P}_d(\boldsymbol{y}_{d-1}) - \boldsymbol{y}_d||_2,$$
$$\text{s.t. } \boldsymbol{\delta}_d^* = \arg\max_{||\boldsymbol{\delta}_d|| < \eta} \mathcal{L}_{adv}(\boldsymbol{y}_d + \boldsymbol{\delta}_d, \boldsymbol{x}_d), \tag{1}$$

where $d \in [D - l + 1, D]$ and $\eta$ is a scale hyperparameter.

As illustrated in Figure 2, we follow a coarse-to-fine strategy, where average pooling is used for downsampling, and classical tensor network (CTN) is applied at coarse scales. This pipeline effectively suppresses adversarial perturbations in the early stages of reconstruction. Specifically, for an adversarial example $\boldsymbol{x}_{adv}$, its downsampled version is denoted as $\boldsymbol{x}_{adv}^{D-l}$, where $\boldsymbol{x}_{adv}^{D-l} \approx \boldsymbol{x}_{cln}^{D-l} + \Delta, \Delta \sim \mathcal{N}(0, \sigma^2)$. At this stage, minimizing the traditional $\ell_2$ loss is sufficient to remove such noise and mitigate the impact of adversarial perturbations, obtaining the clean output. However, as the resolution increases, the distribution of perturbations begins to deviate from normality. This poses a new issue during reconstruction, where minimizing the traditional $\ell_2$ loss will inadvertently restore the perturbations. As a result, the reconstructed example $\boldsymbol{y}$ tends to collapse back toward the adversarial example $\boldsymbol{x}_{adv}$, rather than approximating the unobserved clean example $\boldsymbol{x}_{cln}$.

To suppress the reconstruction of adversarial perturbations on finer scales, we introduce an auxiliary variable $\boldsymbol{\delta}^*$, optimized via an inner maximization process that utilizes a non-convex loss function $\mathcal{L}_{adv}$. This variable serves as the discrepancy between adversarial and clean examples, thereby helping prevent inadvertently restoring the perturbations. We employ a perceptual metric, structural similarity index measure (SSIM, Hore & Ziou, 2010), as $\mathcal{L}_{adv}$ to explore more potential solutions and better

handle complex perturbation patterns. While $\boldsymbol{\delta}^*$ does not exactly represent the true perturbation, bounding $\|\boldsymbol{\delta}\| < \eta$ can help ensure that the misalignment between $\boldsymbol{y}$ and $\boldsymbol{x}_{adv}$ remains controlled, effectively preventing $\boldsymbol{y}$ from simply collapsing into the adversarial example $\boldsymbol{x}_{adv}$.

However, precisely because $\boldsymbol{\delta}^*$ does not represent the true perturbation, minimizing $||\boldsymbol{x}_d - (\boldsymbol{y}_d + \boldsymbol{\delta}_d^*)||_2$ may not yield the desired clean example. To address this issue, we introduce a second loss term $||\mathrm{P}_d(\boldsymbol{y}_{d-1}) - \boldsymbol{y}_d||_2$, which serves as a surrogate prior. Specifically, we utilize the reconstructed downsampled example $\boldsymbol{y}_{d-1}$ as an additional constraint to aid in approximating the $\boldsymbol{x}_{cln}$. Building upon the observations in Figure 1, we start from the resolution $D - l$ example $\boldsymbol{x}_{D-l}$ that is optimized by CTN, and then perform upsampling to the higher resolution to produce a clean-leaning reference, which acts to nudge $\boldsymbol{y}$ toward a less perturbed distribution. Although we never have direct access to the true clean example $\boldsymbol{x}_{cln}$, our loss provides an effective surrogate prior and guides the optimization process. Notably, this two-term optimization neither requires explicit modeling of the perturbation nor knowledge of the attack, allowing TNP to generalize effectively across diverse adversarial scenarios. We provide additional analysis and discussion in Appendix E.6, and the detailed algorithm of our adversarial optimization process is presented in Algorithm 1.

# 5 EXPERIMENTS

In this section, we conduct comprehensive experiments on multiple datasets across various settings. The results demonstrate that TNP achieves robustness with strong generalization. In addition, we present visual comparisons and ablation experiments to further support and explain our contributions.

## 5.1 EXPERIMENTAL SETUP

**Datasets and model architectures**  We conduct extensive experiments on CIFAR-10, CIFAR-100 (Krizhevsky et al., 2009), and ImageNet (Deng et al., 2009) to empirically validate the effectiveness of the proposed methods against adversarial attacks. For classification tasks, we utilize the pre-trained ResNet (He et al., 2016) and WideResNet (Zagoruyko & Komodakis, 2016) models.

**Adversarial attacks**  We evaluate our method against AutoAttack (Croce & Hein, 2020), a widely used benchmark that integrates both white-box and black-box attacks. Additionally, following the guidance of Lee & Kim (2023), we utilize PGD (Madry et al., 2018) with EOT (Athalye et al., 2018b) for a more comprehensive evaluation. Considering the potential robustness overestimation caused by obfuscated gradients of the purifier model, we utilize BPDA (Athalye et al., 2018a) as an adaptive attack with the knowledge of both purifier and classifier, following the setting by Yang et al. (2019); Lin et al. (2024). Further implementation details and discussion are provided in Appendix C.

**Compared methods**  We conduct experiments on the common benchmark and compare the robustness of our method with those listed in RobustBench (Croce et al., 2021). We evaluate the generalization of existing defense methods, including AT methods (Gowal et al., 2020; 2021; Laidlaw et al., 2021; Dolatabadi et al., 2022; Pang et al., 2022) and AP methods, with particular attention to diffusion-based AP (Yoon et al., 2021; Nie et al., 2022; Lee & Kim, 2023; Lin et al., 2025). Furthermore, we include comparisons with Tensor Train (TT, Oseledets, 2011), Tensor Ring (TR, Zhao et al., 2016), quantized technique (Khoromskij, 2011) and PuTT (Loeschcke et al., 2024).

Due to the high computational cost of evaluating methods with multiple attacks, following the guidance of Nie et al. (2022), we randomly select 512 images from the test set for robust evaluation. All experiments presented in the paper are conducted by NVIDIA RTX A5000 with 24GB GPU memory, CUDA v11.7, and cuDNN v8.5.0 in PyTorch v1.13.11. More details in Appendix D.

## 5.2 ROBUSTNESS COMPARISON ON ROBUSTBENCH

We evaluate our method for defending against AutoAttack and compare it with the methods under all adversarial settings listed in RobustBench (Croce et al., 2021). Tables 1 to 4 present the performance of various defense methods against $l_\infty$ and $l_2$ threats. Overall, the highest robust accuracy achievable by our method is generally on par with existing methods without using extra data introduced by Carmon et al. (2019). Specifically, compared to the second-best method, our method improves the robust accuracy by 1.67% on CIFAR-100, by 1.84% on ImageNet, and the average robust accuracy by 0.36% on CIFAR-10. In addition, to ensure a fair comparison, we also evaluate diffusion-based

Table 1: Standard and robust accuracy (%) against AutoAttack $l_\infty$ threat ($\epsilon = 8/255$) on CIFAR-10. [†] use synthetic images. [*] use a robust classifier.

| Defense method | Extra data | Standard Acc. | Robust Acc. |
|---|---|---|---|
| Gowal et al. (2020) | ✓ | 89.48 | 62.70 |
| Bai et al. (2023) | ✓[†] | 95.23 | 68.06 |
| Chen & Lee (2024) | ✗ | 86.10 | 58.09 |
| Cui et al. (2024) | ✗[†] | 92.16 | 67.73 |
| Nie et al. (2022) | ✗ | 89.02 | 70.64 |
| Zhang et al. (2024) | ✗ | 90.04 | 73.05 |
| Lin et al. (2024) | ✗ | 90.62 | 72.85 |
| Sun et al. (2025) | ✗ | 93.29 | 66.94 |
| Ours | ✗ | 82.23 | 55.27 |
| Ours[*] | ✗ | 91.99 | 72.85 |

Table 2: Standard and robust accuracy (%) against AutoAttack $l_2$ threat ($\epsilon = 0.5$) on CIFAR-10.

| Defense method | Extra data | Standard Acc. | Robust Acc. |
|---|---|---|---|
| Augustin et al. (2020) | ✓ | 92.23 | 77.93 |
| Gowal et al. (2020) | ✓ | 94.74 | 80.53 |
| Wang et al. (2023) | ✗[†] | 95.16 | 83.68 |
| Rebuffi et al. (2021) | ✗[†] | 91.79 | 78.32 |
| Ding et al. (2019) | ✗ | 88.02 | 67.77 |
| Nie et al. (2022) | ✗ | 91.03 | 78.58 |
| Zollicoffer et al. (2025) | ✗ | 85.40 | 77.90 |
| Ours | ✗ | 82.23 | 68.16 |
| Ours[*] | ✗ | 91.99 | 79.49 |

Table 3: Standard and robust accuracy (%) against AutoAttack $l_\infty$ ($\epsilon = 8/255$) on CIFAR-100.

| Defense method | Extra data | Standard Acc. | Robust Acc. |
|---|---|---|---|
| Hendrycks et al. (2019) | ✓ | 59.23 | 28.42 |
| Debenedetti et al. (2023) | ✓ | 70.76 | 35.08 |
| Cui et al. (2024) | ✗[†] | 73.85 | 39.18 |
| Wang et al. (2023) | ✗[†] | 75.22 | 42.67 |
| Pang et al. (2022) | ✗ | 63.66 | 31.08 |
| Jia et al. (2022) | ✗ | 67.31 | 31.91 |
| Ours | ✗ | 62.30 | 44.34 |

Table 4: Standard and robust accuracy (%) against AutoAttack $l_\infty$ threat ($\epsilon = 4/255$) on ImageNet.

| Defense method | Extra data | Standard Acc. | Robust Acc. |
|---|---|---|---|
| Salman et al. (2020) | ✗ | 64.02 | 37.89 |
| Bai et al. (2021) | ✗ | 67.38 | 35.51 |
| Nie et al. (2022) | ✗ | 67.79 | 40.93 |
| Bai et al. (2024) | ✗ | 70.41 | 41.70 |
| Chen & Lee (2024) | ✗ | 68.76 | 40.60 |
| Ours | ✗ | 65.43 | 42.77 |

AP using a robust classifier, and TNP still achieves better performance, as shown in Table 12. These results are consistent across multiple datasets and norm threats, preliminarily confirming the effectiveness of our method and its potential for defending against adversarial attacks.

## 5.3 Generalization comparison across diverse adversarial scenarios

As previously highlighted, the existing defense methods are often criticized for their lack of generalization across different norm threats, attacks, and datasets. In the following, we evaluate the performance of our method under diverse adversarial scenarios to demonstrate its robust generalization.

**Results analysis on different norm threats**
Table 5 shows that AT methods (Laidlaw et al., 2021; Dolatabadi et al., 2022) are limited in defending against unseen attacks and can only effectively defend against the specific attacks on which they are trained. An intuitive idea is to apply AT across all norm threats or develop more general constraints to obtain a robust model. However, training such a model is challenging due to the inherent differences among various attacks. In contrast, AP methods (Nie et al., 2022; Lin et al., 2024) exhibit strong generalization, effectively defending against unseen attacks. The results demonstrate that our method also possesses strong generalization capabilities against unseen attacks, achieving performance close to the other AP methods while significantly outperforming the existing AT methods. Specifically, compared to the best AT method, our method improves average robust accuracy by 26.45%.

Table 5: Standard and robust accuracy (%) against AutoAttack $l_\infty$ ($\epsilon = 8/255$) and $l_2$ ($\epsilon = 1.0$) threats on CIFAR-10 with ResNet-50.

| Type | Defense method | SA | Robust Acc. | |
|---|---|---|---|---|
| | | | AA $l_\infty$ | AA $l_2$ |
| | Standard Training | 94.8 | 0.0 | 0.0 |
| AT | Training with $l_\infty$ | 86.8 | 49.0 | 19.2 |
| | Training with $l_2$ | 85.0 | 39.5 | 47.8 |
| | Laidlaw et al. (2021) | 82.4 | 30.2 | 34.9 |
| | Dolatabadi et al. (2022) | 83.2 | 40.0 | 33.9 |
| AP | Nie et al. (2022) | 88.2 | 70.0 | 70.9 |
| | Lin et al. (2024) | 89.1 | 71.2 | 73.4 |
| | Ours | 88.3 | 73.2 | 67.0 |

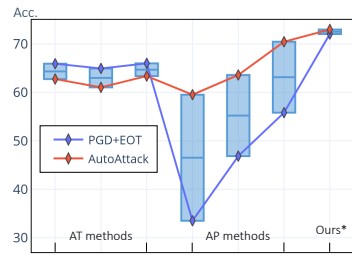

Figure 3: Comparison of robust accuracy against multiple attacks.

Table 6: Standard accuracy (SA) and robust accuracy (RA) against AutoAttack $l_\infty$ ($\epsilon = 8/255$) on CIFAR-10 and CIFAR-100. The pre-trained generative model used in AP is trained on CIFAR-10.

| Method | CIFAR-10 | | CIFAR-100 | | Avg. | |
|---|---|---|---|---|---|---|
| | SA | RA | SA | RA | SA | RA |
| Standard | 94.78 | 0.00 | 81.86 | 0.00 | 88.32 | 0.00 |
| AT | 92.16 | 67.73 | 73.85 | 39.18 | 83.01 | 53.46 |
| AP | 89.02 | 70.64 | 38.09 | 33.79 | 63.56 | 52.22 |
| Ours* | 91.99 | 72.85 | 71.48 | 44.53 | 81.74 | 58.69 |

**Results analysis on different attacks** Figure 3 shows the comparison of robust accuracy against PGD+EOT and AutoAttack with $l_\infty$ ($\epsilon = 8/255$) on CIFAR-10 with WideResNet-28-10. When facing different attacks within the same threat, AT methods (Gowal et al., 2020; 2021; Pang et al., 2022) exhibit better generalization than AP methods (Yoon et al., 2021; Nie et al., 2022; Lee & Kim, 2023). Typically, robustness evaluation is based on the worst-case results of the robust accuracy. Under this criterion, our method outperforms all AT and AP methods. Specifically, compared to the best AP method, our method improves average robust accuracy by 9.39%.

**Results analysis on different datasets** Table 6 shows the generalization of the methods (Nie et al., 2022; Cui et al., 2024) across different datasets. As previously highlighted, the existing AP methods typically rely on specific datasets. For AP method, when a pre-trained generative model trained on CIFAR-10 is applied to adversarial robustness evaluation on CIFAR-100, both standard accuracy and robust accuracy drop significantly. This occurs because the pre-trained generative model can only generate the data it has learned. Although the input examples originate from CIFAR-100, the generative model attempts to output one of the ten classes from CIFAR-10, severely distorting the semantic information of the input examples and leading to low classification accuracy. In contrast, our method exhibits strong generalization across different datasets, achieving comparable robust performance on CIFAR-100 as on CIFAR-10. Specifically, compared to the diffusion-based AP method (Nie et al., 2022), our method improves the average robust accuracy by 6.47%.

**Remark** Unlike existing methods, TNP employs an optimization-based strategy that operates solely on the given input information, without relying on prior knowledge learned from large-scale training datasets or strong assumptions about attacks. This enables TNP to retain strong generalization across diverse adversarial scenarios.

## 5.4 WHY TENSOR NETWORK PURIFICATION WORKS

To understand the contribution of Tensor Network Purification (TNP), we conduct ablation studies and quantitative analysis, with more comparisons provided in Appendix E. We further include a discussion of the limitations and potential directions for future work.

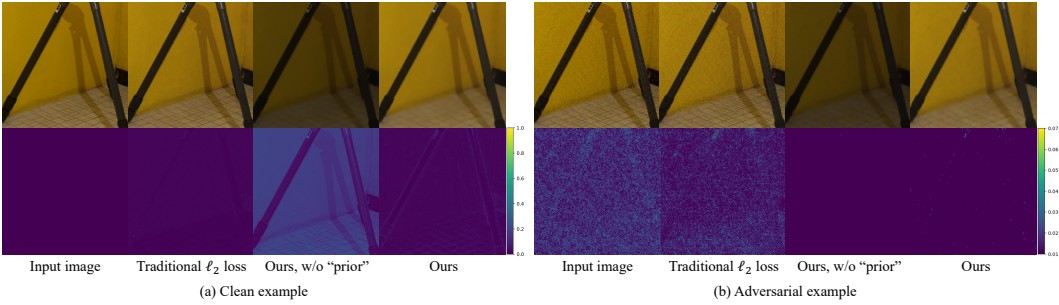

Figure 4: Comparison of visualizations. Top: original image and three reconstructed versions for clean example (CE) and adversarial example (AE). Bottom: error maps between (a) reconstructed clean and original clean examples, and (b) reconstructed adversarial and reconstructed clean examples.

Table 7: Performance comparison of various TN methods on CIFAR-10, evaluated using accuracy, NRMSE, SSIM, and PSNR on reconstructed clean and adversarial examples.

| Defense | Clean example | | | | Adversarial example | | | | Rec. consistency | | |
| method | Acc. | NRMSE | SSIM | PSNR | Acc. | NRMSE | SSIM | PSNR | NRMSE | SSIM | PSNR |
|---|---|---|---|---|---|---|---|---|---|---|---|
| Standard | 94.78 | - | - | - | 0.00 | - | - | - | - | - | - |
| TT | 87.30 | 0.0507 | 0.9526 | 31.14 | 36.13 | 0.0650 | 0.8977 | 28.99 | 0.0267 | 0.9790 | 39.10 |
| TR | **94.34** | **0.0171** | **0.9938** | **40.58** | 0.98 | **0.0464** | **0.9210** | **31.91** | 0.0322 | 0.9598 | 35.51 |
| QTT | 84.57 | 0.0613 | 0.9253 | 29.49 | 51.56 | 0.0724 | 0.8808 | 28.06 | 0.0233 | 0.9855 | 39.88 |
| QTR | 83.40 | 0.0613 | 0.9254 | 29.49 | 49.41 | 0.0724 | 0.8785 | 28.06 | 0.0231 | 0.9853 | 39.96 |
| PuTT | 80.86 | 0.0626 | 0.9261 | 29.32 | 44.14 | 0.0742 | 0.8787 | 27.84 | 0.0311 | 0.9770 | 38.03 |
| Ours | 82.23 | 0.0644 | 0.9203 | 29.06 | **55.27** | 0.0748 | 0.8707 | 27.77 | **0.0218** | **0.9863** | **40.37** |

**Ablation study** Figure 4 shows the comparison of the visualization on ImageNet. For the classical tensor network within the traditional $\ell_2$ loss, we can observe that the reconstructed clean example is almost identical to the original clean example. However, in the presence of adversarial attacks, the same reconstruction behavior inadvertently leads to the preservation of adversarial perturbations, as highlighted in Figure 4b. In contrast, our method better suppresses those perturbations, ensuring that the reconstructed AE and the reconstructed CE retain similar information. Moreover, we evaluate the necessity of the second term in Eq. (1), which serves as a surrogate prior constraint to optimize the reconstructed examples toward the clean data distribution. As shown in Figure 4a, removing this constraint eliminates prior information from the optimization, leading to a significant semantic shift in the reconstructed image. In contrast, applying the full loss function more effectively guides the optimization, suppressing the perturbations while preserving the semantic information. The detailed accuracy results corresponding to these cases are provided and discussed in Appendix E.5.

Furthermore, we conduct additional ablation studies to analyze the impact of downsampling, adversarial optimization objective, and hyperparameter $\eta$ on the robustness of our method, as quantified by the accuracy metrics reported in Tables 14 to 16. These results consistently demonstrate that our method significantly outperforms classical TN methods in removing adversarial perturbations.

**Reconstruction and robustness analysis** Table 7 shows the quantitative results of the reconstruction performance with detailed descriptions of the evaluation metrics provided in Appendix D.2. The reconstructed example is expected to closely match the clean example (CE), while remaining sufficiently different from the adversarial example (AE) to avoid restoring adversarial perturbations. TR achieves the best reconstruction quality for both CE and AE, with a standard accuracy of 94.34%. However, its robust accuracy is only 0.98%, indicating the perturbations are clearly restored during reconstruction. In contrast, TNP achieves the highest robustness with a robust accuracy of 55.27%, while still maintaining reasonable reconstruction quality. Notably, we further evaluate the consistency between reconstructed CE and reconstructed AE, as shown in the "Rec. consistency" column of Table 7. TNP achieves the best performance across all metrics, indicating that the reconstructed CE and reconstructed AE retain highly similar information, suggesting that the perturbations are effectively suppressed. These findings align well with the visual observations in Figure 4, reinforcing the effectiveness of TNP and highlighting its potential in adversarial scenarios.

**Limitations and future works** We identify several open problems related to TNP: (1) Although TNP requires no training cost, its inference cost still leaves room for improvement. We report a comparison of inference times in Table 10, with detailed discussion provided in Appendices E.1 and E.2. (2) As an optimization-based technique, TNP is inherently more resistant to adaptive attacks, see more discussion in Appendix C. Accordingly, developing more advanced optimization strategies and adaptive attacks specifically tailored to TNP remains a valuable direction for future research.

## 6 CONCLUSION

In this paper, we propose Tensor Network Purification (TNP), a novel optimization-based AP framework built upon a specially designed tensor network decomposition. Unlike existing methods that rely heavily on training data or pre-trained generative models, TNP requires zero training cost and operates solely on the given input information. Extensive experiments on CIFAR-10, CIFAR-

100, and ImageNet demonstrate that TNP achieves robust performance, especially exhibiting strong generalization across diverse adversarial scenarios. Additionally, we further identify several open challenges related to TNP, and believe that continued exploration of TN-based AP remains an exciting research direction for developing a plug-and-play and effective defense technique.

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

APPENDIX

## A   INFLUENCE OF DIFFERENT SAMPLING METHODS

To support our hypothesis of using the average pooling, we test it with stride sampling, which selects pixels with constant strides. In principle, the stride sampling would not change the distribution of perturbations. Therefore, it serves as a baseline to compare the influence of distributions.

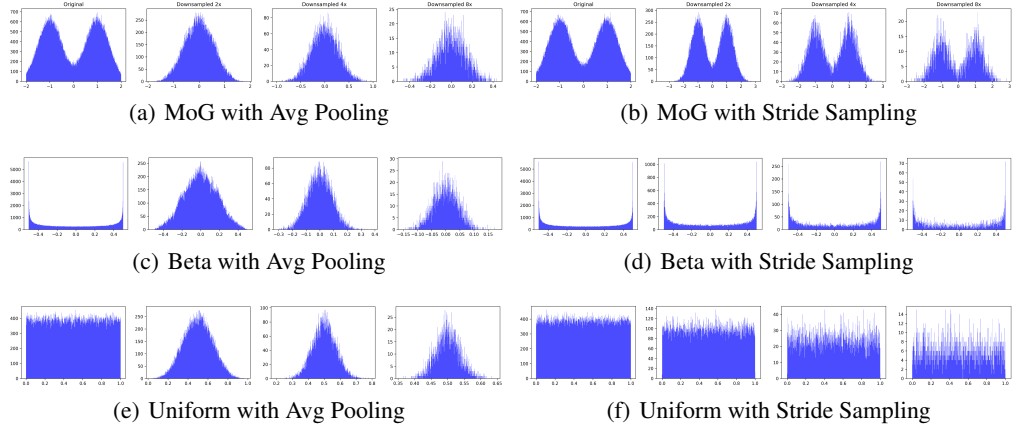

(a) MoG with Avg Pooling          (b) MoG with Stride Sampling

(c) Beta with Avg Pooling          (d) Beta with Stride Sampling

(e) Uniform with Avg Pooling          (f) Uniform with Stride Sampling

Figure 5: Histogram figures of noises under different sampling methods.

We test four types of noise distributions: (1) Gaussian $\mathcal{N}(0, 0.3^2)$, (2) Mixture of Gaussian (MoG), $0.5 \cdot \mathcal{N}(-1.0, 0.5^2) + 0.5 \cdot \mathcal{N}(1.0, 0.5^2)$, (3) Beta distribution, Beta$(0.5, 0.5) - 0.5$, and (4) Uniform distribution, Uniform$(-0.5, 0.5)$. For MoG, Beta and uniform noises, we scale them to have the same signal-to-noise ratio with the Gaussian distribution. We add the noises on the Girl image (Loeschcke et al., 2024) with resolution $1024 \times 1024$. First, we show the noise distributions in Figure 5. As can be seen, the Avg Pooling strategy transforms the non-Gaussian noises into Gaussian-like noises, while the Stride sampling would not. Second, we run the PuTT algorithm with different sampling methods for 100 times. The violin plot of denoising results are shown in Figure 6. In Gaussian distribution, the Stride sampling is better than AvgPooling. While for non-Gaussian noises, the AvgPooling is more robust and better than Stride. The denoising results indicate that the average pooling can handle

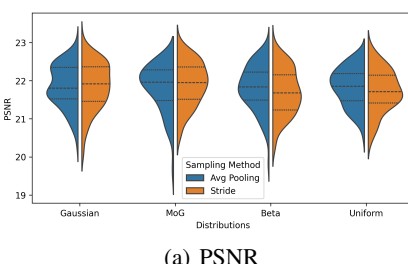 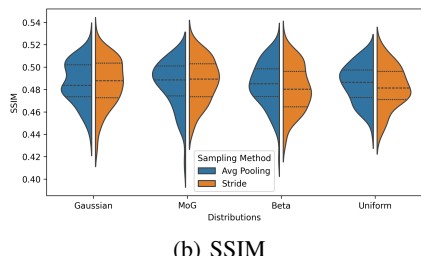

(a) PSNR  (b) SSIM

Figure 6: Violin plot of denoising results using different sampling methods.

different types of noises, which is consistent with our hypothesis. However, as we introduced, this might not be enough, since we need to deal with the original image and noises in the final stage.

## B    TENSOR NETWORK DECOMPOSITION

### B.1    MATRIX PRODUCT OPERATORS

A matrix product operator (MPO, McCulloch, 2008; Hubig et al., 2017) is the TN representation of a linear operator acting on a TT format, which makes it highly efficient to handle large operators. Namely, a linear operator $\boldsymbol{\mathcal{A}} : \mathbb{R}^{I_1 \times \dots \times I_D} \to \mathbb{R}^{J_1 \times \dots \times J_D}$. Namely, if $\boldsymbol{\mathcal{Y}} = \boldsymbol{\mathcal{A}} \boldsymbol{\mathcal{X}}$, then each entry of $\boldsymbol{\mathcal{Y}}$ is given as

$$y_{\mathbf{i}} = \sum_{i_1=1}^{I_1} \cdots \sum_{i_D=1}^{I_D} \boldsymbol{A}^1_{j_1,i_1} \boldsymbol{A}^2_{j_2,i_2} \dots \boldsymbol{A}^D_{j_D,i_D} \boldsymbol{X}^1_{i_1} \boldsymbol{X}^2_{i_2} \dots \boldsymbol{X}^D_{i_D} \,,$$

### B.2    PROLONGATION OPERATOR

This work uses a specific MPO, known as the prolongation operator $\boldsymbol{\mathcal{P}}_d$ (Lubasch et al., 2018), to upsample a QTT format of an image from resolution $d - 1$ to $d$.

Consider a one-dimensional vector $\boldsymbol{x}_d \in \mathbb{R}^{2^d}$. The matrix $\boldsymbol{P}_{2^d \to 2^{d+1}}$ upsamples $\boldsymbol{x}_d$ to $\boldsymbol{x}_{d+1}$ by linear interpolation between adjacent points. For example, for $d = 2$,

$$\boldsymbol{P}_{4 \to 8} = \begin{bmatrix} 1 & 0 & 0 & 0 \\ 0.5 & 0.5 & 0 & 0 \\ 0 & 1 & 0 & 0 \\ 0 & 0.5 & 0.5 & 0 \\ 0 & 0 & 1 & 0 \\ 0 & 0 & 0.5 & 0.5 \\ 0 & 0 & 0 & 1 \\ 0 & 0 & 0 & 0.5 \end{bmatrix}$$

The matrix $\boldsymbol{P}_{2^d \to 2^{d+1}}$ can be written as an MPO $\boldsymbol{\mathcal{P}}_{d+1}$ entry-wise

$$p_{j_1,\dots,j_d,i_1,\dots,i_{d+1}} = \boldsymbol{P}^1_{j_1,i_1} \dots \boldsymbol{P}^d_{j_d,i_d} \boldsymbol{P}^{d+1}_{i_{d+1}} \,.$$

The entries are given explicitly (Lubasch et al., 2018) as

$$\boldsymbol{P}^l_{1,1}(1,1) = \boldsymbol{P}^l_{2,2}(1,1) = \boldsymbol{P}^l_{2,1}(1,2) = \boldsymbol{P}^l_{1,2}(2,2) = 1, \forall l \in [d]$$

$$\boldsymbol{P}^{d+1}_1(1) = 1, \boldsymbol{P}^{d+1}_2(1) = \boldsymbol{P}^{d+1}_2(2) = 0.5 \,,$$

and other entries are zero.

The prolongation operator described above applies to the QTT format of one-dimensional vectors. In general, this operator is the tensor product of the one-dimensional operators on each dimension: $\boldsymbol{\mathcal{P}}^{(2)}_d = \boldsymbol{\mathcal{P}}_d \otimes \boldsymbol{\mathcal{P}}_d$ for 2-dimensions (images) and $\boldsymbol{\mathcal{P}}^{(3)}_d = \boldsymbol{\mathcal{P}}_d \otimes \boldsymbol{\mathcal{P}}_d \otimes \boldsymbol{\mathcal{P}}_d$ for 3-dimensions (3D objects). For simplicity, since this work concerns only images, the superscript is omitted, denoting the prolongation operator as $\boldsymbol{\mathcal{P}}_d$.

Ultimately, for a resolution $d$ image $\boldsymbol{x}_d$, and $\boldsymbol{\mathcal{X}}_d = \mathrm{Q}(\boldsymbol{x}_d)$, the upsampled image is resolution $d + 1$, given as $\mathrm{P}_d(\boldsymbol{x}_d) = \mathrm{Q}^{-1}(\boldsymbol{\mathcal{P}}_d \boldsymbol{\mathcal{X}}_d)$, where the linear function $\mathrm{P}_d(\cdot)$ acts on the image level.

## B.3 RECAP OF PuTT

A $2^D \times 2^D$ image, denoted as $\boldsymbol{x}_D$, can be quantized in to a $D$th order tensor $\boldsymbol{\mathcal{X}}_D = \mathrm{Q}(\boldsymbol{x}_D)$. Firstly, $\boldsymbol{x}_D$ is downsampled by average pooling to $\boldsymbol{x}_{D-l}$, correspondingly possesing a quantization $\boldsymbol{\mathcal{X}}_{D-l}$. Then, $D - l$ QTT cores of $X_{D-l}$ can be optimized by backpropagation, returning $\boldsymbol{\mathcal{Y}}_{D-l}$. The QTT cores of next resolution $\boldsymbol{\mathcal{X}}_{D-l+1}$ can be optimized similarly, initialized by the prologation $\boldsymbol{\mathcal{P}}_{D-l+1}(\boldsymbol{y}_{D-l})$. Repeat the process until the original resolution. (Loeschcke et al., 2024) demonstrates impressive reconstruction capability of PuTT thanks to the QTT structure and coarse-to-fine approach. The pseudocode is given in Algorithm 2.

---

**Algorithm 2** PuTT (Loeschcke et al., 2024)

---

    **Input:** Image $\boldsymbol{x}_D$, number of iterations $T$, upsampling iterations $(t_1, \ldots, t_l)$.
    **Output:** TT reconstruction $\boldsymbol{y}_D = \mathrm{PuTT}(\boldsymbol{x}_D)$.
    $d \leftarrow D - l, \boldsymbol{x}_d \leftarrow \mathrm{AvgPool}(\boldsymbol{x}_D), \boldsymbol{\mathcal{X}}_d \leftarrow \mathrm{Q}(\boldsymbol{x}_d)$
    **for** $t = 1 \rightarrow T$ **do**
        **if** $t \in (t_1, \ldots, t_l)$ **then**
            $d \leftarrow d + 1$
            $\boldsymbol{x}_d \leftarrow \mathrm{AvgPool}(\boldsymbol{x}_D)$
            $\boldsymbol{\mathcal{X}}_d \leftarrow \mathrm{Q}(\boldsymbol{x}_d)$
        **end if**
        Loss $\ell \leftarrow \mathrm{MSE}(\boldsymbol{\mathcal{Y}}_d - \boldsymbol{\mathcal{X}}_d)$
        Update QTT cores $\boldsymbol{\mathcal{Y}}_d$ by backpropagation
    **end for**
    **return** $\boldsymbol{y}_D = \mathrm{Q}^{-1}(\boldsymbol{\mathcal{Y}}_D)$

---

However, while PuTT aims to obtain better initialization by downsampling for better optimization and reconstruction, it does not account for adversarial examples or analyze the impact of downsampling on perturbations. Additionally, PuTT also minimizes the reconstruction loss on the input image, which inevitably results in the reconstruction of the perturbations. In contrast, we focus on the perturbations and propose a new optimization process introduced in the next section, aiming to reconstruct clean examples.

## C IMPLEMENTATION DETAILS OF ADVERSARIAL ATTACKS

We evaluate our method of defending against AutoAttack (Croce & Hein, 2020) and compare with the methods as listed RobustBench benchmark (https://robustbench.github.io). For a comprehensive evaluation, we conduct experiments under all adversarial attack settings. Specifically, we set $\epsilon = 8/255$ and $\epsilon = 0.5/1.0$ for AutoAttack $l_{\inf}$ and AutoAttack $l_2$ threats on CIFAR-10. On CIFAR-100, we set $\epsilon = 8/255$ for AutoAttack $l_{\inf}$. On ImageNet, we set $\epsilon = 4/255$ for AutoAttack $l_{\inf}$. We evaluate our method of defending against PGD+EOT (Madry et al., 2018; Athalye et al., 2018b) and present the comparisons of AT methods, AP methods, and our method. Following the guidelines of (Lee & Kim, 2023), we set $\epsilon = 8/255$ for PGD+EOT $l_{\inf}$ threats on CIFAR-10, where the update iterations of PGD is 200 with 20 EOT samples.

Considering the potential robustness overestimation (Athalye et al., 2018a) caused by obfuscated gradients of purifier model, we utilize BPDA as an adaptive attack (Tramer et al., 2020; Croce et al., 2022), following the setting by (Yang et al., 2019; Lin et al., 2024), which treats the purification step as an identity mapping during the backward pass, effectively bypassing its effect when computing gradients. In all experiments, the attacker has knowledge of both the purifier (TNP) and the classifier (Cls). The target of the attack is a new model $F$, i.e., $F(x) = Cls(TNP(x))$. The reason we chose BPDA is that the existing full gradient attacks are not applicable in TN-based AP due to the memory explosion issues associated with attacking TN optimization. In contrast to diffusion-based AP (Nie et al., 2022; Li et al., 2025; Liu et al., 2025), TN is a model-free technique that does not rely on a fixed model or any parameters for gradient computation. Additionally, the iterative process in TN is a gradual optimization procedure, rather than the fixed inference iterations employed in diffusion-based methods, resulting in surrogate attacks that are difficult to apply to TN. Therefore, we empirically validated the effectiveness of our method through the existing adaptive attacks, e.g., BPDA.

Remark: Unlike conventional AP methods that rely on a specific trained model for purification, TNP is a model-free technique without any parameters or the static network architecture for gradient computation, which is an inference-time optimization strategy. Additionally, the iterative process in TNP is a dynamic, gradual optimization procedure, in contrast to the fixed-step inference in DiffPure. This dynamic nature further hinders the applicability of the gradient checkpointing technique, as there is no static computational graph or predetermined set of parameters to track and store during intermediate steps. In other words, there is no well-defined checkpoint for storing intermediate gradients, thus the gradient checkpointing technique cannot be directly applied to TNP. This is also an inherent advantage of TN-based AP, which significantly increases the difficulty of developing adaptive attacks against TNP. Our paper is the first work to introduce a model-free optimization based method. We look forward that, building on the foundation established in this work, future research will explore adaptive attack strategies specifically tailored to TN-based AP, thereby advancing and refining the defense mechanisms of TN-based AP methods.

## D  MORE DETAILS OF EXPERIMENTAL SETTINGS

### D.1  IMPLEMENTATION DETAILS OF OUR METHOD

For CIFAR-10, CIFAR-100 with resolution $32 \times 32$ and ImageNet with resolution $224 \times 224$, we first upsample them into resolution $2^D \times 2^D$ image $x_D$. Based on the initial experimental results, we set $D = 8$, $l = 1$, $\alpha = 0.1$, initial $\beta = 0.008$, $N = 1$, and the number of optimization iterations is 2048 for the following experiments. For the scale hyperparameter $\eta$, we set $\eta = 0.1$ in all our experiments without knowing the specific attack norm. Since adversarial perturbations are very small, a fixed $\eta = 0.1$ already exceeds the scale of most attacks. Moreover, choosing a larger $\eta$ can introduce excessive noise, leading to lower-quality reconstructions. Based on our preliminary experiments, $\eta = 0.1$ offers a suitable balance and thus serves as our default setting. The table results presented in the paper are conducted under these hyperparameters. This trick creates a large enough image to downsample until the perturbations are well mixed into Gaussian noise. Furthermore, without this initial step, the semantic information can become almost indistinguishable after several downsampling steps, especially for low-resolution images. For example, if a $32 \times 32$ image is reduced with the factor of 8, the resolution $4 \times 4$ image is of poor quality. Additionally, to more clearly observe the denoising effects in visualization results, we upsample the images to resolution $D = 11$ with $\alpha = 0.05$, $\eta = 0.1$ and $N = 3$ for the experiments in Figure 4, and comparisons in different downsampled images in Figure 1. The code will be available upon acceptance, with details provided in the configuration files.

### D.2  IMPLEMENTATION DETAILS OF EVALUATION METRICS

We evaluate the performance of defense methods using multiple metrics: Standard accuracy and robust accuracy (Szegedy et al., 2014) on classification tasks. For denoising tasks, we measure the Normalized Root Mean Squared Error (NRMSE, Botchkarev, 2018), Structural Similarity Index Measure (SSIM, Hore & Ziou, 2010), Peak Signal-to-Noise Ratio (PSNR) metrics between a reference image $\boldsymbol{x}$ and its reconstruction $\boldsymbol{y}$, where pixel values are in $[0, 1]$.

Normalized Root Mean Squared Error

$$\text{NRMSE}(\boldsymbol{x}, \boldsymbol{y}) = \frac{\|\boldsymbol{x} - \boldsymbol{y}\|_2}{\|\boldsymbol{x}\|_2} = \frac{\sqrt{\sum_i (\boldsymbol{x}_i - \boldsymbol{y}_i)^2}}{\sqrt{\sum_i \boldsymbol{x}_i^2}}.$$

Structural Similarity Index Measure

$$\text{SSIM}(\boldsymbol{x}, \boldsymbol{y}) = \frac{(2\mu_x \mu_y + C_1)(2\sigma_{xy} + C_2)}{(\mu_x^2 + \mu_y^2 + C_1)(\sigma_x^2 + \sigma_y^2 + C_2)},$$

where: $\mu_x$ and $\mu_y$ are the mean pixel values of images $\boldsymbol{x}$ and $\boldsymbol{y}$. $\sigma_x^2$ and $\sigma_y^2$ are the variances of $\boldsymbol{x}$ and $\boldsymbol{y}$. $\sigma_{xy}$ is the covariance between $\boldsymbol{x}$ and $\boldsymbol{y}$. $C_1$ and $C_2$ are small constants to stabilize the division.

Peak Signal-to-Noise Ratio

$$\text{PSNR}(\boldsymbol{x}, \boldsymbol{y}) = 10 \log_{10} \left( \frac{1}{\text{MSE}(\boldsymbol{x}, \boldsymbol{y})} \right).$$

NRMSE, SSIM and PSNR evaluate reconstructed image quality from error, structural-similarity, and signal-to-noise perspectives, making them particularly suitable and comprehensive for assessing reconstruction performance. In traditional denoising and reconstruction tasks, generally a lower NRMSE, a higher SSIM, and a higher PSNR generally indicate better performance.

# E    COMPARISONS

## E.1    ADVERSARIAL DEFENSE METHODS

Table 8: Comparison of defenses with vanilla model on CIFAR-10 (negative impacts are marked in red and positive impacts are marked in green). #: Using pre-trained diffusion model. Unseen datasets: Applying the model trained on CIFAR-10 to CIFAR-100 evaluation.

| Defense method | Clean examples | Adv. examples | Unseen attacks | Unseen datasets | Training costs | Inference costs |
|---|---|---|---|---|---|---|
| Vanilla model | ~95% | ~0% | ~0% | ~0% | 0 | ~0 |
| Expectation | ≈ | ↑↑ | ↑↑ | ↑↑ | 0 | ~0 |
| AT | ↓↓ | ↑↑ | N/A | N/A | ↑↑ | ~0 |
| AP# | ↓ | ↑↑ | ↑↑ | N/A | ↑↑ | ↑↑ |
| TNP (ours) | ↓ | ↑↑ | ↑↑ | ↑↑ | 0 | ↑ |

In the development of adversarial defense methods, with the emergence of adversarial attacks, numerous methods have been proposed, including adversarial training (AT) and adversarial purification (AP). As research in this area progresses, researchers have gradually moved beyond defenses tailored to specific attacks and begun developing more general defense techniques that enhance model robustness and generalization against unseen attacks and datasets.

As mentioned before, AT predominantly consists of retraining the model on a finite set of adversarial examples, thereby conferring robustness primarily against those known perturbations. However, this process closely resembles a form of overfitting: the classifier becomes highly specialized to the attack patterns learned during training, at the expense of its performance on clean examples. As a result, standard accuracy typically degrades, and the robustness to withstand previously unseen attacks remains severely limited, as shown in Table 5.

Another class of defense methods is AP, which leverages pre-trained generative models trained on clean examples, thus can effectively defend against all types of attacks. However, AP is constrained by the specific dataset used during training, making it difficult to transfer effectively to new tasks or data distributions. As shown in Table 6, when applying the diffusion model trained on CIFAR-10 to CIFAR-100 evaluation, the standard accuracy dropped by 35.76% compared with AT.

Therefore, both mainstream defense methods face significant generalization challenges. To address this, one possible solution is to re-train the robust classifier to defend against new attacks or train a new generator on new datasets. However, such strategies incur substantial computational overhead and training costs, making them impractical for deployment in adversarial environments characterized by continuously emerging attacks, as summarized in Table 8.

To tackle these challenges with AT and AP, we propose a novel defense technique based on tensor network representation, which eliminates the need for training a powerful generative model or relying on specific dataset distributions, making it a general-purpose adversarial purification. In the experiments, TNP has shown great advantages in these challenges: 26.45% improvement in average

Table 9: The standard accuracy, robust accuracy, and inference time on CIFAR-10.

| Methods | Standard Accuracy | Robust Accuracy | Inference Time |
|---|---|---|---|
| TNP | 91.99 | 72.85 | 2.45 s |
| TNP-L | 89.45 | 71.68 | 0.49 s |

robust accuracy over AT across different norm threats; 9.39% improvement over AP across multiple attacks; 6.47% improvement over AP across different datasets. Furthermore, to further improve inference efficiency, we introduce a lightweight variant on CIFAR-10 with a robust classifier (Cui et al., 2024), named TNP-L, which reduces the number of optimization iterations to 768, substantially decreasing the inference time, as shown in Table 9. Remarkably, TNP achieves these benefits with zero additional training cost, offering an efficient solution for adversarial purification.

## E.2 INFERENCE TIME COST

Table 10 shows the inference time of different methods on CIFAR-10 and ImageNet, which is measured on a single image. We leverage the parallelization to further improve the computational efficiency of TNP and conducted experiments on a single GPU.

Table 10: Comparison of per-image inference time on a single GPU.

| Methods | CIFAR-10 | ImageNet | Avg. |
|---|---|---|---|
| DiffPure (Nie et al., 2022) | 1.49 s | 5.11 s | 3.30 s |
| GDMP (Wang et al., 2022) | 5.17 s | 11.27 s | 8.22 s |
| Lee & Kim (2023) | 14.90 s | 35.09 s | 25.00 s |
| SSNI (Sun et al., 2025) | 4.47 s | 14.44 s | 9.46 s |
| AGDM (Lin et al., 2025) | 1.73 s | 5.52 s | 3.63 s |
| TNP (Ours) | 2.45 s | 3.13 s | 2.79 s |
| TNP-L (Ours) | 0.49 s | 3.13 s | 1.81 s |

Specifically, TNP requires 2.45 seconds per image on CIFAR-10, outperforming most diffusion-based approaches such as GDMP, SSNI, and Lee & Kim (2023). To further alleviate the inference burden, we reduce the number of optimization iterations on CIFAR-10, which led to a substantial reduction in inference time to 0.49 seconds. The corresponding accuracy is reported in Table 9. On ImageNet, TNP achieves the lowest inference time of 3.13 seconds among all compared methods. These results highlight the strong potential of TNP for high-resolution image purification. Notably, in addition to inference speed, TNP achieves robust generalization without any training or fine-tuning, which saves substantial training resources compared to AT and AP methods, as discussed in Appendix E.1. This enables TNP to flexibly deploy across diverse datasets and attacks, making it suitable for diverse adversarial scenarios.

In summary, the inference time of TNP is already lower than that of standard diffusion-based AP methods. We believe there is still room for further improvement, making TNP more suitable for real-world deployment. Accordingly, we leave the integration of tensor network-based AP techniques with more advanced and faster optimization strategies as an open problem for future work.

## E.3 ZERO-SHOT ADVERSARIAL DEFENSE

Table 11: Comparison with untrained networks against AutoAttack $l_\infty$ ($\epsilon = 8/255$) on CIFAR-10.

| Defense method | Acc. | NRMSE | SSIM | PSNR |
|---|---|---|---|---|
| Clean examples | | | | |
| DIP | 90.43 | 0.0464 | 0.9565 | 32.13 |
| MAE | 88.28 | 0.0847 | 0.8842 | 26.90 |
| Ours | 82.23 | 0.0644 | 0.9203 | 29.06 |
| Adversarial examples | | | | |
| DIP | 38.28 | 0.0451 | 0.9467 | 32.53 |
| MAE | 1.56 | 0.0914 | 0.8472 | 26.24 |
| Ours | 55.27 | 0.0748 | 0.8707 | 27.77 |

AT and AP methods depend heavily on the external training dataset, overlooking the potential internal priors in the input itself. Among adversarial defense techniques, untrained neural networks such as deep image prior (DIP, Ulyanov et al., 2018) and masked autoencoder (MAE, He et al., 2022)

have been utilized to avoid the need of extra training data (Dai et al., 2020; 2022; Lyu et al., 2023). However, although such deep learning models achieve high-quality reconstruction results, they have been shown to be susceptible to revive also the adversarial noise. This section compares two representative untrained models DIP and MAE.

Table 11 shows that although DIP and MAE have achieved remarkable standard accuracy and reconstruction quality, they deteriorate significantly under attack.

### E.4 MORE EXPERIMENTS

Due to the overfitting of WideResNet-28-10 trained on the limited data available in CIFAR-10, we observe that the results with standard classifier struggle to reach competitive performance, consistent with findings from Chen & Lee (2024). To further improve robust accuracy, we conduct experiments with the robust classifier, which utilizes an additional 20M synthetic images in training (Cui et al., 2024). This leads to a significant improvement in robust accuracy on CIFAR-10. Moreover, compared to the used robust classifier (Cui et al., 2024), our method further improves the robust accuracy by 5.12%. To ensure a fair and consistent comparison, we also consider employing a robust classifier for diffusion-based AP method in Table 12. Using a robust classifier on CIFAR-10 for diffusion-based AP leads to a slight improvement in robust accuracy. Meanwhile, our method with AT consistently maintains better performance.

Table 12: Standard accuracy and robust accuracy on CIFAR-10.

| Defense method | Standard Acc. | Robust Acc. |
|---|---|---|
| Strandard | 94.78 | 0.00 |
| AT | 92.16 | 67.73 |
| DiffPure | 89.02 | 70.64 |
| DiffPure + AT | 90.76 | 71.68 |
| Ours + AT | 91.99 | 72.85 |

Table 13: Standard accuracy and robust accuracy against PGD+EOT ($l_\infty$, $\epsilon = 8/255$) on CIFAR-10.

| Type | Defense method | Standard Acc. | Robust Acc. |
|---|---|---|---|
| Adv. Training | (Pang et al., 2022) | 88.62 | 64.95 |
| | (Gowal et al., 2020) | 88.54 | 65.93 |
| | (Gowal et al., 2021) | 87.51 | 66.01 |
| DM-based AP | (Yoon et al., 2021) | 85.66 | 33.48 |
| | (Nie et al., 2022) | 91.41 | 46.84 |
| | (Lee & Kim, 2023) | 90.16 | 55.82 |
| | (Lin et al., 2025) | 90.42 | 64.06 |
| | Ours* | 91.99 | 72.07 |

Recently, Lee & Kim (2023) conducted a thorough investigation and proposed a robust evaluation guideline using PGD+EOT. To undertake a more comprehensive evaluation, we further evaluate our method following the guidelines in this part. Table 13 shows the results on CIFAR-10, and the observations are basically consistent with the existing experiments, supporting our method as a powerful defense technique and more effective than existing AT or AP methods.

### E.5 ABLATION STUDY

We investigate how downsampling, adversarial optimization, hyperparameter $\eta$ affect the robustness of our tensor network-based purification.

Table 14: Ablation study on the effectiveness of downsampling (DS) and adversarial optimization objective (L1: introducing the auxiliary variable $\delta^*$; L2: introducing the second loss term).

| DS | L1 | L2 | SA | RA | Avg. |
|---|---|---|---|---|---|
| ✗ | ✗ | ✗ | 87.30 | 36.13 | 61.72 |
| ✓ | ✗ | ✗ | 80.86 | 44.14 | 62.50 |
| ✓ | ✓ | ✗ | 66.41 | 50.59 | 58.50 |
| ✓ | ✗ | ✓ | 82.61 | 52.53 | 67.57 |
| ✓ | ✓ | ✓ | 82.23 | 55.27 | 68.75 |

Table 14 presents an ablation study evaluating the impact of downsampling (DS) and the adversarial optimization objectives including: L1, introducing the auxiliary variable $\delta^*$; L2, introducing the

second loss term on the robustness of our tensor network-based purification method against AutoAttack ($l_\infty = 8/255$.) on CIFAR-10. Without DS or adversarial optimization objective (first row), the method achieves a standard accuracy (SA) of 87.30% but only robust accuracy (RA) of 36.13%. Introducing DS alone (second row) improves RA to 44.14%, indicating that downsampling helps mitigate adversarial perturbations. Based on this, adding only L1 (third row) further increases RA to 50.59% but at the cost of a significant drop in SA to 66.41%. This is consistent with the observations in Figure 4, where without L2 serving as a surrogate prior, it may lead to increasing the likelihood of significant deviation in the wrong direction. On the other hand, incorporating only L2 (fourth row) achieves a better balance, with SA at 82.61% and RA at 52.53%. Finally, combining DS with both L1 and L2 (last row) yields the best overall performance, with SA of 82.23%, RA of 55.27%, and the highest average accuracy. This demonstrates that both downsampling and the full adversarial optimization objective are essential for robustness while maintaining high standard accuracy.

| Table 15: AutoAttack $l_\infty$ on CIFAR-10. | | | | | Table 16: AutoAttack $l_\infty$ on ImageNet. | | | | |
|---|---|---|---|---|---|---|---|---|---|

| $\eta$ | 0 | 0.1 | 0.2 | 0.3 | | $\eta$ | 0 | 0.1 | 0.2 | 0.3 |
|---|---|---|---|---|---|---|---|---|---|---|
| SA | 82.61 | 82.23 | 63.67 | 49.21 | | SA | 60.93 | 65.43 | 27.73 | 19.72 |
| RA | 52.53 | 55.27 | 42.57 | 30.66 | | RA | 39.26 | 42.77 | 24.80 | 18.55 |
| Avg. | 67.57 | 68.75 | 53.12 | 39.94 | | Avg. | 50.10 | 54.10 | 26.27 | 19.14 |

We further analyze the impact of the hyperparameter $\eta$ in the adversarial optimization objective, as shown in Tables 15 and 16. As mentioned in Appendix D.1, as $\eta$ increases further, both SA and RA drop significantly. This is because excessive $\eta$ introduces too much noise, which degrades the reconstruction quality and harms both standard and robust performance. In summary, a moderate value of $\eta$ (e.g., 0.1) provides the best trade-off between robustness and clean accuracy, while excessively large values lead to degraded performance. This observation supports our choice of setting $\eta = 0.1$ as the default across all experiments. Notably, by using the same hyperparameters across different attack types, norm threats, and perturbation scales, we achieve consistently strong performance under diverse adversarial settings, further demonstrating the generalization of our method.

### E.6 MORE ANALYSIS AND DISCUSSION

**Analysis on Tensor Netwokr Purification (TNP):** Initially, TNP is motivated by observations derived from downsampling and the central limit theorem. As detailed in Section 4.1 and Appendix A, the downsampling using average pooling can effectively transform adversarial perturbations into a normal-like distribution at coarse scales. Specifically, for an adversarial example, the downsampled version is denoted as $x'_{D-l}$, where $x'_{D-l} \approx x_{D-l} + \Delta, \Delta \sim \mathcal{N}(0, \sigma^2)$. At this stage, minimizing the traditional $l_2$-norm loss ($||x - y||_2$) can remove such noise and mitigate the impact of adversarial perturbations, obtaining the clean version output $y_{D-l}$, where $y_{D-l} \approx x_{D-l}$.

However, as the resolution increases, the distribution of perturbations diverges from normality, and minimizing the $l_2$-norm loss inadvertently leads to restoring the perturbations at fine scales. Consequently, the reconstruction $y$ tends to collapse back toward the adversarial example $x'$, rather than approximating the unobserved clean example $x$. To avoid this issue, we introduce $\delta^*$ into the optimization objective. This additional variable allows the optimization to allocate adversarial perturbations to $\delta^*$ instead of forcing $y$ to absorb them entirely, thereby preventing $y$ from collapsing into $x'$. Nevertheless, since $\delta^*$ does not represent the true perturbation, minimizing $||x - (y + \delta^*)||_2$ alone may not yield the desired clean example. Therefore, we further introduce a second loss term $||P_d(y_{d-1}) - y_d||_2$, which serves as a "surrogate prior". Intuitively, the coarse-scale reconstruction $y_{D-l}$ is already a clean version and can guide the optimization process, pushing the higher-resolution output $y_d$ toward a less perturbed distribution. Importantly, this two-term optimization neither requires explicit modeling of the perturbation nor knowledge of the attack, allowing TNP to generalize effectively across diverse adversarial scenarios.

**Discussion:** As we all know, adversarial challenge of attack and defense is endless. This contradiction arises from the fundamental difference between adversarial attacks and defenses. Attacks are inherently destructive, whereas defenses are protective. This adversarial relationship places the attacker in an active position, while the defender remains passive. As a result, attackers can continually explore new attack strategies against a fixed model to degrade its predictive performance, ultimately

leading to the failure of conventional defenses. The introduction of TNP has the potential to address this issue. As a model-free technique, TNP generates representations solely based on the input information. These representations dynamically change with each input, preventing attackers from exploiting a fixed model to generate effective adversarial examples. This defensive mechanism allows TNP to maintain a more proactive stance in the ongoing competition between attacks and defenses.

## F    HISTOGRAM AND KERNEL DENSITY ESTIMATION RESULTS

Figure 7 shows the histogram and kernel density estimation of adversarial perturbations on 10 images. The distribution of those perturbations progressively aligns with that of Gaussian noise as the downsampling process progresses.

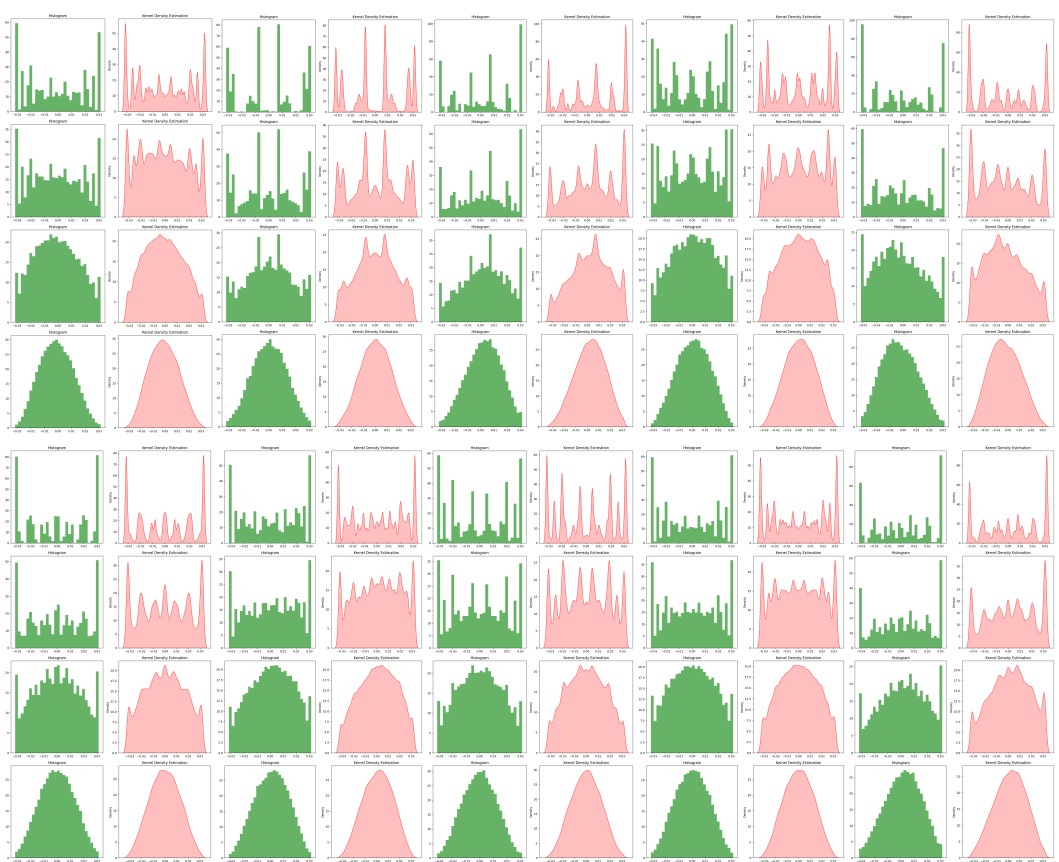

Figure 7: The histogram and kernel density estimation of perturbations in the downsampled images.

## G    LLM USAGE DISCLOSURE

In accordance with the latest ICLR policy regarding the use of large language models (LLMs), we clarify that LLMs were utilized solely for grammar correction and language refinement of the manuscript. No LLMs were involved in the conceptualization, research design, implementation, experimentation, or any other scientific aspects of this work.

## H    VISUALIZATION

We visualize low-resolution examples from CIFAR-10 and high-resolution examples from ImageNet, as shown in Figures 8 and 9.

Clean examples

Adversarial examples

Reconstructed examples

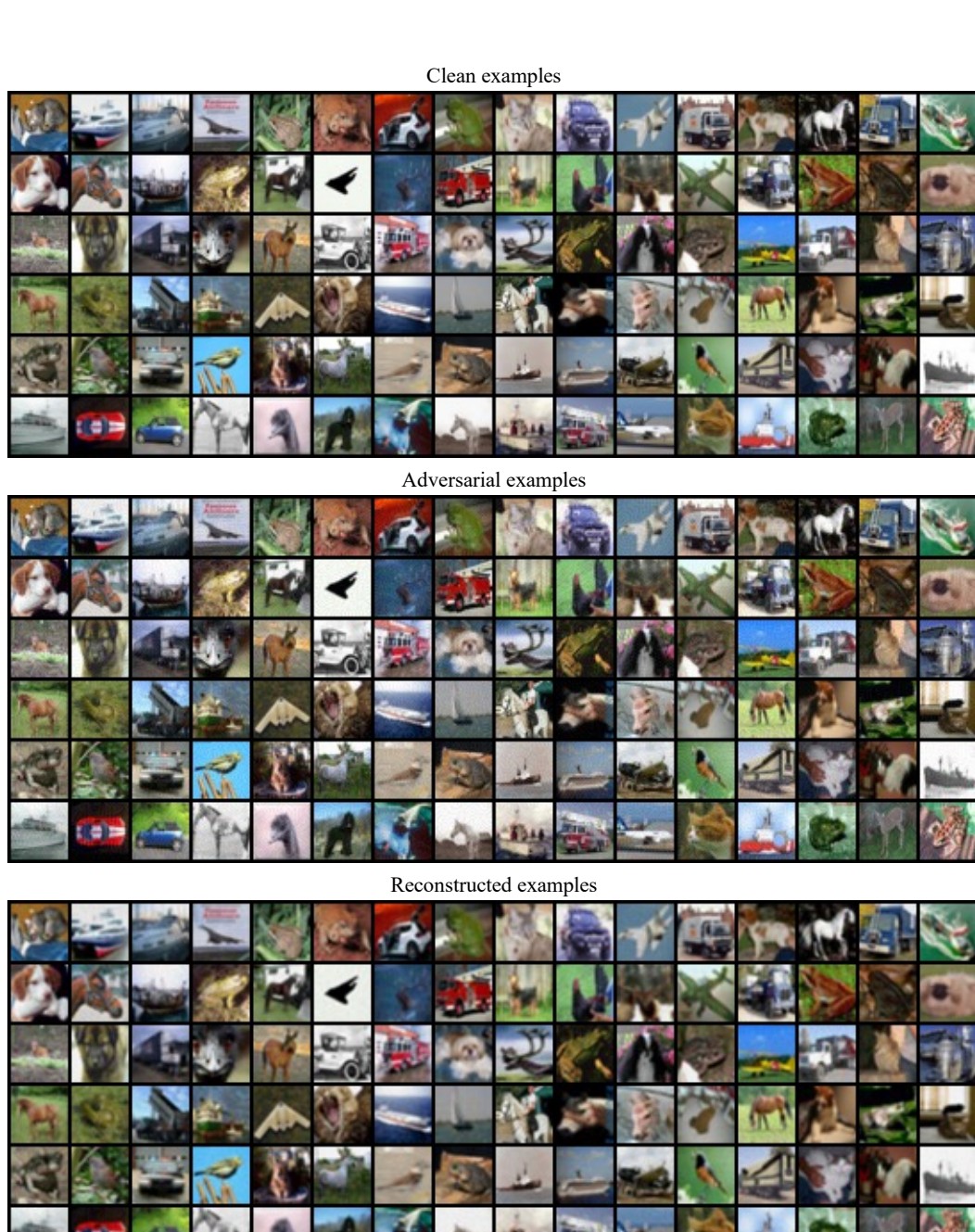

Figure 8: Clean examples (Top), adversarial examples (Middle) and reconstructed examples (Bottom) of CIFAR-10.

Clean examples

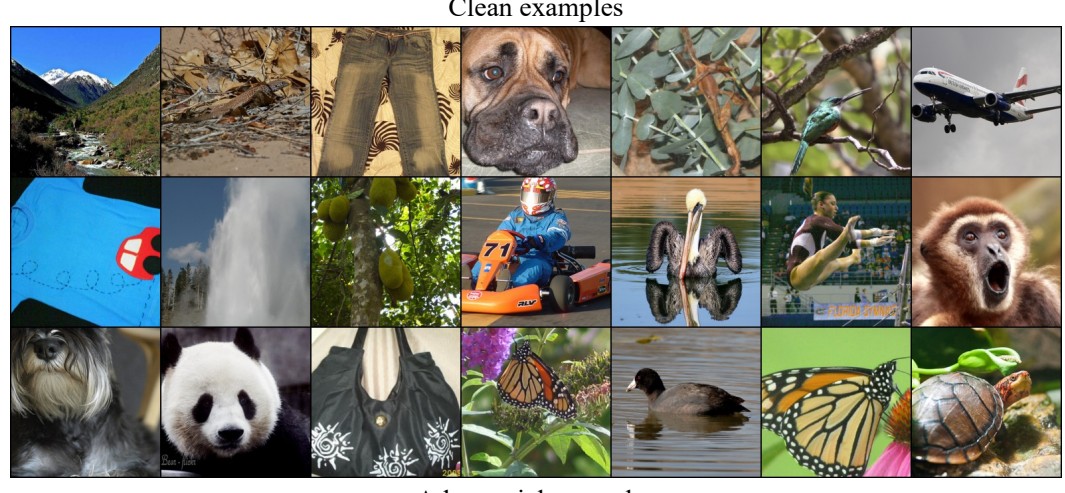

Adversarial examples

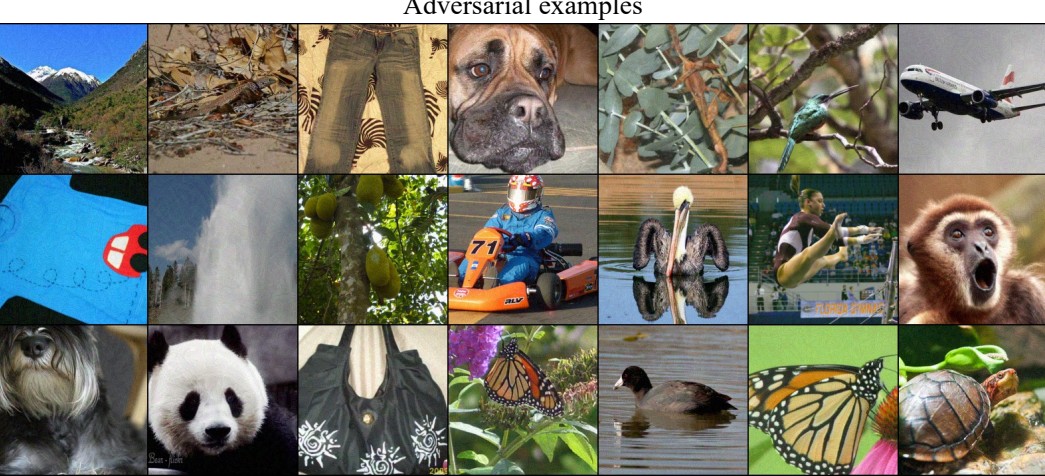

Reconstructed examples

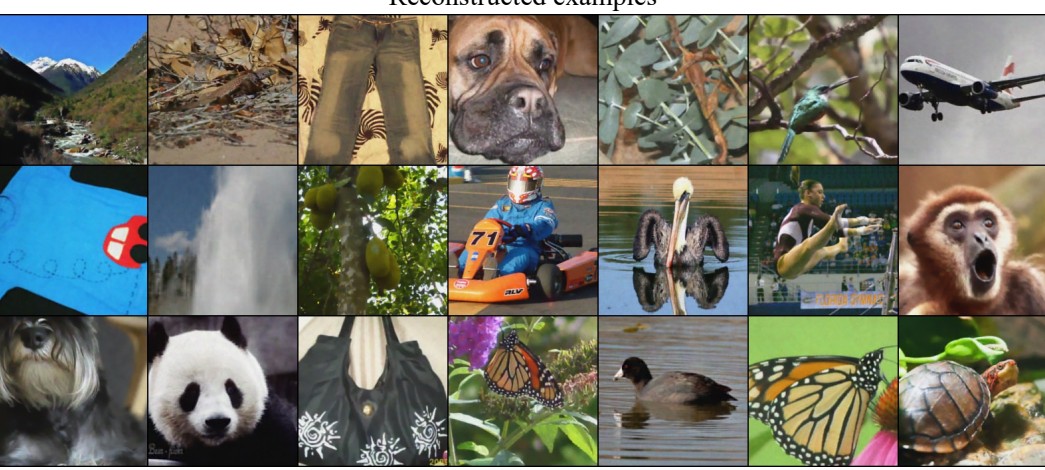

Figure 9: Clean examples (Top), adversarial examples (Middle) and reconstructed examples (Bottom) of ImageNet.

