# OpenReview forum: "Optimization-Based Defender via Coarse-To-Fine Tensor Network Representation"
_ICLR.cc/2026/Conference — Submitted to ICLR 2026_

### Official Review · Reviewer_zp4E · 2025-10-29

**Soundness:** 3
**Presentation:** 4
**Contribution:** 4
**Rating:** 8
**Confidence:** 4

**Summary:**

This paper introduces an AP method combined a coarse-to-fine TN incremental learning algorithm with an adversarial optimization objective to defend adversarial attacks by mitigating adversarial perturbations. It’s training-free and shows its generalizable performance in various models and datasets and compared with other SOTA level AT and diffusion-based AP methods.

**Strengths:**

1. This paper introduces a novel method with a TN and dexterously combined with an adversarial optimization objective which effectively address the gaps in AT, AP and TN, shows the SOTA level performance and it’s training free.
2. The theory is presented clearly, and the theory diagrams are also clear.
3. The experiments are concise and comprehensive with various models and datasets under various attacks. Providing some visual results for making more convinced. And the compared methods are up-to-date.

**Weaknesses:**

1. There’s no results w/o auxiliary variable in Figure 4. And there has comprehensive comparison in Table 14 but in appendix.
2. The author chose the hyperparameter $\eta$ in Table 15 and Table 16. However, due to the significant performance differences among the 0, 0.1 and 0.2, the experimental granularity is not sufficient to convince readers that 0.1 is the relatively better choice.
3. The experiment has not been run multiple times, and there is no report of the mean and standard deviation of the results.

**Questions:**

1. Could you add a result w/o auxiliary variable in Figure 4, or integrate the comprehensive comparison into the main text rather than appendix.
2. Could you conduct a more granular search within the range of 0 to 0.2 to present the trend of performance with respect to $\eta$ more precisely to verify the current selection. You can also add CIFAR-100 and create a curve graph to show the same trend across different datasets.

---

> ### Author Response · Authors · 2025-11-19
> **Responses to Reviewer zp4E**
>
> We sincerely appreciate Reviewer zp4E for your time and constructive feedback. We are pleased that the Reviewer recognizes the **novelty**, **presentation quality**, and **experimental comprehensiveness** of our work. The following are our responses to the comments and questions raised.
>
>
>
> > **Weakness 1 \& Question 1:** There’s no results w/o auxiliary variable in Figure 4. And there has comprehensive comparison in Table 14 but in appendix. Could you add a result w/o auxiliary variable in Figure 4, or integrate the comprehensive comparison into the main text rather than appendix.
>
> **Response:** We will add the result w/o auxiliary variable to Figure 4. In addition, we believe that integrating the comprehensive comparison into the main text can further improve the readability of the manuscript. Therefore, we will also move Table 14 to the main text. These modifications will be incorporated into the camera-ready version.
>
> > **Weakness 2 \& Question 2:** Due to the significant performance differences among the 0, 0.1 and 0.2 in Table 15 and Table 16, the experimental granularity is not sufficient to convince readers that 0.1 is the relatively better choice. Could you conduct a more granular search over this hyperparameter? You can also add CIFAR-100 and create a curve graph to show the same trend across different datasets.
>
> **Response:** We appreciate the Reviewer's helpful suggestions. We have conducted an initial evaluation of the hyperparameter $\eta$ on CIFAR-10 using a finer granularity of 0.02, as shown in the table below. In addition, we will extend this experiment with a finer-grained search (e.g., step size 0.01) on CIFAR-10, CIFAR-100, and ImageNet in the camera-ready version. We will also convert the corresponding tables into curve graphs to present the results in a more intuitive manner.
>
> |||||||||||||
> |:-:|:-:|:-:|:-:|:-:|:-:|:-:|:-:|:-:|:-:|:-:|:-:|
> |$\eta$| 0.00 | 0.02 | 0.04 | 0.06 | 0.08 | 0.10 | 0.12 | 0.14 | 0.16 | 0.18 | 0.20 |
> | SA | 82.61 | 81.84 | 81.84 | 81.45 | 81.05 | 82.23 | 77.14 | 72.27 | 69.92 | 66.02 | 63.67 |
> | RA | 52.53 | 53.71 | 54.10 | 51.17 | 53.90 | 55.27 | 52.54 | 53.51 | 50.98 | 49.60 | 42.57 |
> | Avg. | 67.57 | 67.78 | 67.97 | 66.31 | 67.48 | 68.75 | 64.84 | 62.89 | 60.45 | 57.81 | 53.12 |
> ||||||
>
>
> > **Weakness 3:** The experiment has not been run multiple times, and there is no report of the mean and standard deviation of the results.
>
> **Response:** We have run the experiments an additional five times and report the mean and standard deviation of the results on CIFAR-10 against AutoAttack, as shown in the table below. We will include these results, along with more experiments under other settings, in the camera-ready version.
>
> |||||||||
> |:-:|:-:|:-:|:-:|:-:|:-:|:-:|:-:|
> | | Org.   | 1     | 2     | 3     | 4     | 5     |  Mean $\pm$ Std |
> | SA |  82.23 | 82.81 | 83.20 | 82.42 | 83.01 | 82.03 | 82.62 $\pm$ 0.46 |
> | RA |  55.27 | 54.88 | 55.66 | 54.68 | 55.85 | 56.05  | 55.40 $\pm$ 0.55 |
> |||||||||
>
>
> We sincerely appreciate the Reviewer's insightful comments and suggestions, which provide several valuable directions for improving the quality of the manuscript. We will incorporate all of the above modifications and the mentioned additional experiments into the camera-ready version to further enhance the comprehensiveness of our work.

---

> > ### Comment · Reviewer_zp4E · 2025-11-25
> >
> > Thanks for this response and the additional experiments. Most of my concerns have been addressed, I will maintain my initial high rating.

---

> > > ### Author Response · Authors · 2025-11-26
> > >
> > > Dear Reviewer zp4E,
> > >
> > > We sincerely appreciate your follow-up feedback. We are pleased that our responses have addressed most of your concerns. Thank you again for your time, support, constructive comments, and for maintaining your high rating.

---

### Official Review · Reviewer_xrYF · 2025-10-31

**Soundness:** 2
**Presentation:** 1
**Contribution:** 3
**Rating:** 4
**Confidence:** 3

**Summary:**

The paper proposes Tensor Network Purification (TNP), a defense mechanism designed to reconstruct clean examples from their adversarial perturbations. TNP works by reconstructing progressively downsampled versions of the perturbed input, via a tensor network (TN) decomposition in conjunction with a novel objective that suppresses the reinstatement of the perturbations. The intuition behind this approach is that downsampling with average pooling smooths the perturbative noise towards a normal distribution, which can be effectively handled by the TN decomposition. Unlike previous methods, TNP does not depend on a pre-trained model or a specific dataset, and works as an input pre-process step for any classifier, which does not need to be retrained. The authors perform extensive experimentation, showcasing the effectiveness of the proposed mechanism with respect to various threats and attack types on CIFAR-10, CIFAR-100 and ImageNet.

**Strengths:**

1.	The experiments and ablation study performed by the authors are very extensive and yield especially good results.
2.	The proposed mechanism is not susceptible to some of the pitfalls of previously proposed adversarial training and adversarial purification methodologies.
3.	The authors provide a clear discussion of TNP’s limitations, which strengthens the paper's contribution and provides clear avenues for future work.

**Weaknesses:**

1.	A major problem with the current manuscript is that it is somewhat poorly written. First of all, there are various grammatical and syntactical mistakes, as well as typos. For example, see p.1 l.52 “which also implies that allowing for robust defense without retraining the classifier”, p.4 l.211 “we define the linear function ... acts on the image level”, and the title of Section 3 “Backgrounds”.
2.	Even though loosely discussed during the introduction, the general setting and an explicit definition of the problem TNP aims to address are not apparent until halfway through Section 4. As a result, the reader is forced to discover both the method and the problem it is aimed at concurrently. It would have been much smoother had these been formalized before Section 4.
3.	The discussion in Subsection 4.1 is either problematic or calls for additional explanation. In particular, an important assumption of the Central Limit Theorem is that the random variables are independent and identically distributed. The case is somewhat similar in the conditional setting that is referenced. However, since a specific setting is not formalized, it is unclear why this assumption is met. In the same spirit, in Appendix A, average pooling is contrasted with stride sampling so as to support its choice. This is not sufficient support. First, demonstrating the superiority of the method of choice versus a single other alternative does not constitute a principled justification. Second, because both of these are not properly formalized, the results are difficult to actually validate.

**Questions:**

1.	Can you elaborate on the choice of tensor network decompositions instead of a different kind of model, like a small neural network fitted on a single data point, for example? Have you experimented in this direction? Why do tensor networks deal with perturbative noise that follows the normal distribution?
2.	Can you further explain the reasoning of Subsection 4.1? What assumptions are being made on the distribution of the noise (see the second point in the Weaknesses section)? For example, is it only on the pixel level? Moreover, another point to be made here is that the CLT suggests that an appropriate sequence of random variables converges to a normal distribution. Are the downsampling steps enough to justify invoking the CLT?

---

> ### Author Response · Authors · 2025-11-19
> **Responses to Reviewer xrYF (1/3)**
>
> We sincerely appreciate Reviewer xrYF for your time and constructive feedback. We are pleased that the Reviewer recognizes the value of our **experiments**, **ablation studies**, **robust mechanism**, and **discussion of limitations**. The following are our responses to the comments and questions raised.
>
> > **Weakness 1:** A major problem with the current manuscript is that it is somewhat poorly written. First of all, there are various grammatical and syntactical mistakes, as well as typos. For example, see p.1 l.52 ''which also implies that allowing for robust defense without retraining the classifier'', p.4 l.211 ''we define the linear function ... acts on the image level'', and the title of Section 3 ''Backgrounds''.
>
> **Response:** We appreciate the Reviewer's careful reading and for pointing out these issues. We have thoroughly proofread the manuscript in the revised version, including but not limited to the specific grammatical errors you identified (e.g., p.1 l.52; p.4 l.211) and the update of the section title from ''Backgrounds'' to ''Background''.
>
> > **Weakness 2:** Even though loosely discussed during the introduction, the general setting and an explicit definition of the problem TNP aims to address are not apparent until halfway through Section 4. As a result, the reader is forced to discover both the method and the problem it is aimed at concurrently. It would have been much smoother had these been formalized before Section 4.
>
> **Response:** Thank you for this helpful suggestion. We fully agree that introducing the general setting and clearly defining the problem earlier would improve the readability of the paper. Accordingly, we have revised the third and fourth paragraphs in the Introduction section, as detailed below.
>
> p.1 l.59: ''... Serving as a plug-and-play pre-processing step, TN-based AP can eliminate potential adversarial perturbations from both clean and adversarial examples before they are fed into the classifier (Yoon et al., 2021). This implies that robust defense can be achieved without retraining the classifier model. Specifically, an input example is first transformed into a higher-order tensor representation, which is then decomposed into a set of low-rank tensor cores through TN decomposition. By optimizing these tensor cores under a specially designed constraint objective, the TN decomposition effectively suppresses the undesirable perturbations while maintaining the essential semantic content of the image. Finally, the purified image is reconstructed from the optimized tensor cores and passed to the standard classifier for prediction.
>
> However, classical TN methods are primarily designed for image completion and denoising tasks in which the corruption is sparse or follows a Gaussian distribution. In contrast, the distribution of adversarial perturbations fundamentally differs from these assumptions and often aligns with the intrinsic statistics of the data (Ilyas et al., 2019; Zhu \& Li, 2022). As a result, these perturbations behave more like genuine features than noise, making them difficult to model explicitly and prone to being inadvertently reconstructed. To address this issue, we ...''
>
> This content will be further refined for improved readability and incorporated into the camera-ready version.

---

> ### Author Response · Authors · 2025-11-19
> **Responses to Reviewer xrYF (2/3)**
>
> > **Weakness 3 \& Question 2:** The discussion in Subsection 4.1 is either problematic or calls for additional explanation. Can you further explain the reasoning of Subsection 4.1?
> a) Since a specific noise setting is not formalized, it is unclear why the assumption is met. What assumptions are being made on the distribution of the noise?
> b) For example, is it only on the pixel level?
> c) Are the downsampling steps enough to justify invoking the CLT?
> d) The comparison between average pooling and stride sampling in Appendix A does not provide sufficient support for the choice of average pooling.
>
> **Response:** We appreciate the Reviewer's insightful comments and questions. We are more than willing to provide further clarification and justification, as outlined below.
>
> a) To illustrate the motivation, we interpret adversarial perturbations through a random field perspective. Specifically, we model the perturbations as a random field, denoted as $\{\Delta_i\}$, where $i \in \mathbb{Z}^2$ is the index of pixels. Under several mild assumptions, the CLT can be applied, and the perturbations become closer to a Gaussian distribution after average pooling than before. We justify these assumptions in more detail below.
>
> - Second-order stationarity. (1) The mean is a constant, e.g., $\mathbb{E}[\Delta_i] = \mu$ for all $i$. In particular, we can assume the perturbation has zero mean. (2) The covariance is only dependent on relative positions, e.g., $Cov(\Delta_i, \Delta_j) = C(i - j)$. This is also a natural assumption for images, and is particularly suitable for CNNs with weight sharing and invariance.
> - Finite moments. In particular, the second moment is finite, e.g., $\mathbb{E}[\Delta^2_i] < \infty$. This assumption is also reasonable since perturbations are bounded in adversarial attacks.
> - Mixing property. There are many types of mixing properties [1]. In general, this assumption requires the correlations to vanish for long-range dependencies. For example, Bolthausen (1982) assumes the $\alpha$-mixing condition [2], e.g., $\alpha(r) \to 0$ as $r \to \infty$, where $\alpha$ is the $\alpha$-mixing coefficient and $r$ is the distance. Again, this local dependency assumption is reasonable for images, and is adopted for many image models like CNN and Autoregressive models.
>
> Under these assumptions, the average $M = \frac{1}{k} \sum_{i \in \mathcal{B}} \Delta_i$ becomes increasingly close to a Gaussian distribution as $k$ increases, where $\mathcal{B}$ is a $k \times k$ block.
> [1] Bulinski and Shashkin. Limit theorems for associated random fields and related systems. World Scientific, 2007.
> [2] Bolthausen. On the central limit theorem for stationary mixing random fields. The Annals of Probability, 1982.
>
> b) Our assumptions rely on the stationary and mixing property of the random field. At the pixel level, we assume bounded variance and constant mean value.
>
> c) Theoretically, the convergence speed of the CLT depends on the above assumptions, especially the mixing property. In our method, we do not rely on an asymptotic convergence statement with infinitely many downsampling steps. Instead, our method uses a fixed number of pooling layers, where the averaging block size $k$ grows exponentially with the pooling depth. We empirically verify that at this coarse scale, the perturbation distribution is already well approximated by a Gaussian. In particular, Figure 1b and Figure 7 in the Appendix show that the KL divergence to a matched Gaussian decreases substantially over the downsampling process.
>
> d) We fully agree that contrasting average pooling with stride sampling alone is not a complete ''principled justification'' of our design choice. Our goal in Appendix A is more modest: To illustrate that different downsampling schemes have qualitatively different effects on the perturbation distribution. Stride sampling simply subsamples pixels and therefore preserves the marginal distribution of the perturbation, while average pooling replaces a block of pixels with their mean. Under the random field model, the former does not change the perturbation distribution, whereas the latter yields the block averages to which CLT behavior applies. This difference is reflected empirically in Appendix A, where average pooling transforms non-Gaussian noises into Gaussian-like distributions and achieves better TN denoising, while stride sampling leaves the non-Gaussian structure largely unchanged.

---

> ### Author Response · Authors · 2025-11-19
> **Responses to Reviewer xrYF (3/3)**
>
> > **Question 1:**
> a) Can you elaborate on the choice of tensor network decompositions instead of a different kind of model, like a small neural network fitted on a single data point, for example?
> b) Have you experimented in this direction?
> c) Why do tensor networks deal with perturbative noise that follows the normal distribution?
>
> **Response:**
> a) Tensor networks (TNs) are classical tools in signal processing and have been widely successful in tasks such as image completion and denoising. These capabilities motivate us to explore their potential for removing adversarial perturbations. More importantly, our choice of TNs is driven by several key factors that align closely with the goals of adversarial purification.
>
> - First, TNs rely entirely on the input data for optimization, allowing them to operate without dependence on specific datasets, pre-trained models, or attack assumptions. This property is well suited for building a generalizable defense.
>
> - Second, there are no fixed model parameters in TNs and the tensor cores are updated throughout the optimization process, which inherently increases the difficulty of adversarial attacks.
>
> - Finally, the multi-scale representation of TNs naturally aligns with our coarse-to-fine optimization strategy. For these reasons, we choose TNs as the foundational architecture for achieving our objective.
>
> b) In the original manuscript, we have experimented with small neural networks fitted on single data points, as shown in Table 11 and discussed in Appendix E.3. These small neural networks achieve high accuracy on clean examples, but their performance deteriorates significantly under adversarial attacks.
>
> c) Tensor networks are particularly effective for perturbative noise that follows a normal distribution [3]. In most standard formulations, the low-rank tensor factors are learned by minimizing an L2 (least-squares) loss, which corresponds to maximum-likelihood estimation under additive Gaussian noise [4], so the resulting low-rank tensor contractions behave as optimal linear filters that suppress small Gaussian perturbations.
> [3] Zhao et al. Bayesian CP factorization of incomplete tensors with automatic rank determination. IEEE transactions on pattern analysis and machine intelligence, 2015.
> [4] Hastie et al. The elements of statistical learning. Springer series in statistics New-York, 2009.

---

### Official Review · Reviewer_2dCC · 2025-10-31

**Soundness:** 3
**Presentation:** 2
**Contribution:** 2
**Rating:** 4
**Confidence:** 4

**Summary:**

This paper introduces TNP, a training-free adversarial defense framework. It addresses the known failure of classical TNs on non-Gaussian adversarial perturbations by hypothesizing that average-pooling downsampling causes these perturbations to approximate a normal distribution. The method proposes a coarse-to-fine optimization pipeline guided by a new adversarial objective, which uses the coarser, cleaner reconstruction as a "surrogate prior" to guide the finer-scale reconstruction. The primary claimed contribution is a data-agnostic purifier that achieves generalization across unseen datasets and attacks compared to existing generative-model-based AP methods.

**Strengths:**

1.  The paper proposes a training-free framework built on tensor network decomposition. The articulated motivation connects average-pooling downsampling to the Central Limit Theorem as a way to handle non-Gaussian adversarial perturbations.
2.  The method is evaluated in a cross-dataset setting (training on CIFAR-10, testing on CIFAR-100).

**Weaknesses:**

1.  The paper's evaluation against adaptive attacks is critically flawed. It relies only on BPDA, a known weak attack proxy. A full EOT evaluation is absent, with "memory explosion issues" cited as the reason. This justification is insufficient and raises concerns about gradient obfuscation.
2.  The method's practical utility is low due to poor efficiency. The reported inference time on CIFAR-10 is 2.45s per image, which is noted as slower than diffusion-based baselines and presents a significant barrier to use.
3.  The paper's SOTA claims are not supported by its own data. On the standard CIFAR-10 $l_\infty$ and $l_2$ benchmarks, the method's performance is demonstrably below or on-par with existing works.

**Questions:**

1.  Regarding the adaptive attack evaluation, could the authors elaborate on the technical barriers (e.g., "memory explosion") that prevented a full EOT evaluation?
2.  The paper reports a 2.45s inference time on CIFAR-10 (Table 10), while the robust accuracy on this dataset appears on-par with prior work. Could the author discuss more about this cost-performance trade-off?

---

> ### Author Response · Authors · 2025-11-19
> **Responses to Reviewer 2dCC (1/2)**
>
> We sincerely appreciate Reviewer 2dCC for your time and constructive feedback. We are pleased that the Reviewer recognizes the **motivation** and **cross-dataset evaluation** of our work. The following are our responses to the comments and questions raised.
>
> > **Weakness 1 \& Question 1:** The paper's evaluation against adaptive attacks is critically flawed, relying only on BPDA without a full EOT evaluation. The ''memory explosion'' explanation is insufficient and raises concerns about gradient obfuscation.
> Could the authors elaborate on the technical barriers (e.g., ''memory explosion'') that prevented a full EOT evaluation?
>
> **Response:** We appreciate the Reviewer's attention to the full EOT evaluation. The main technical barrier to conducting a full EOT evaluation lies in the inherently optimization-based nature of TNP, as discussed in Appendix C of the original manuscript. Below is a clearer version for responding to this comment.
> - One key technical difficulty is that TNP performs a long, instance-specific optimization involving many inference iterations. When combined with iterative attacks such as EOT, the attacker faces a nested iterative optimization issue. This double iteration structure greatly amplifies computational and memory costs, forming one barrier to applying full EOT evaluation to TNP.
> - We fully agree that memory explosion alone is not a sufficient reason to prevent a full EOT evaluation. Similar memory explosion issues also arise in CNN-based AP methods such as diffusion-based purifiers, which likewise involve the multi-step iterations. In those models, gradient checkpointing has proven effective for mitigating this issue by computing gradients in stages while storing checkpoints to recover the final full gradient. However, these techniques are unfortunately inapplicable to TNP for the following reasons.
>
>   - Unlike conventional defense methods that rely on a fixed, parameterized neural network, TNP works in a fundamentally different manner. TNP constructs a tensor network representation and performs instance-specific optimization to purify each example. During this process, TNP contains no learned or fixed model parameters that provide a stable computation graph for gradient computation.
>
>   - In addition, the iterative process in TNP is a dynamic and gradual optimization procedure. The tensor cores are continuously updated during optimization, and each update depends on the evolving intermediate states. This is also fundamentally different from diffusion-based purifiers, whose inference consists of fixed-step iterations over a fixed neural network. The dynamic nature of TNP further prevents a full EOT evaluation
>
>   As a result, due to the above characteristics of TNP, full EOT evaluation cannot be applied to TNP.
>
> We hope this additional explanation helps address your concern, and we will further clarify these technical barriers and challenges in the camera-ready version to improve the overall presentation.

---

> ### Author Response · Authors · 2025-11-19
> **Responses to Reviewer 2dCC (2/2)**
>
> > **Weakness 2:** The method's practical utility is low due to poor efficiency. The reported inference time on CIFAR-10 is 2.45s per image, which is noted as slower than diffusion-based baselines and presents a significant barrier to use.
>
> **Response:** We respectfully note that TNP is more efficient than most diffusion-based baselines, as shown in Table 10. Among five diffusion-based methods evaluated on CIFAR-10 and ImageNet, TNP achieves faster inference than three methods on CIFAR-10 and all methods on ImageNet, demonstrating competitive efficiency. We agree that an inference time of 2.45 seconds per image remains insufficient for practical deployment. To address this limitation, we have further improved TNP and developed a faster variant that requires only 0.49 seconds per image on CIFAR-10, as shown in Table 9 of the original manuscript. Compared with the fastest diffusion-based baseline (1.49 seconds), our method achieves a substantial improvement in efficiency.
>
> > **Weakness 3:** The paper's SOTA claims are not supported by its own data. On the standard CIFAR-10 $l_\infty$ and $l_2$ benchmarks, the method's performance is demonstrably below or on-par with existing works.
>
> **Response:** We respectfully clarify that the primary contribution of our work is to address the challenge of robustness generalization, rather than to achieve state-of-the-art performance under any single adversarial setting. After revisiting the manuscript, we recognize that two instances of the phrase ''state-of-the-art performance'' in the main text may have unintentionally caused confusion. Accordingly, we will revise these descriptions to more accurately reflect our contribution and to avoid misunderstanding.
>
> To further clarify, our focus is on achieving strong robustness generalization across diverse adversarial scenarios. Under this objective, our method demonstrates substantial improvements, as supported by our experimental results. Specifically, TNP achieves a 26.45\% improvement in average robust accuracy over AT across different norm threats and a 9.39\% improvement over AP across multiple attacks. In addition, it achieves an 18.18\% improvement in average standard accuracy and a 6.47\% improvement in average robust accuracy over AP across different datasets.
>
>
> > **Question 2:** The paper reports a 2.45s inference time on CIFAR-10 (Table 10), while the robust accuracy on this dataset appears on-par with prior work. Could the author discuss more about this cost-performance trade-off?
>
> **Response:** Regarding the cost-performance trade-off, we would like to respond to it from two perspectives: computational cost and generalization performance, which also serve as a further summary of our responses to Weaknesses 2 (W2) and 3 (W3).
>
> - As discussed in our response to W2, we agree that an inference time of 2.45 seconds is a lack of practicality. To address this, we further optimized the efficiency of our method and achieved an inference time of 0.49 seconds on CIFAR-10, which is substantially faster than the diffusion-based baselines. In addition, we further discuss the training cost in Appendix E.1. To address generalization challenges, both AT and AP methods require retraining, which typically demands substantial computational resources (e.g., over 10 hours of training for a diffusion model on CIFAR-10). In contrast, TNP is a data-agnostic defense method and requires no training, thereby saving substantial training costs.
>
> - As discussed in our response to W3, the robust accuracy indeed appears on-par with prior work under a single adversarial setting. However, the primary contribution of TNP is its strong robustness generalization. From this perspective, TNP demonstrates clear advantages.
>
> To recap, we propose a new and promising AP direction for robustness generalization: a novel optimization-based AP using tensor networks, with zero training cost and acceptable inference cost, achieving strong robustness generalization across diverse adversarial scenarios.

---

### Official Review · Reviewer_tpP1 · 2025-11-01

**Soundness:** 3
**Presentation:** 3
**Contribution:** 3
**Rating:** 6
**Confidence:** 4

**Summary:**

This paper addresses the limited generalization of adversarial purification defenses by proposing a novel optimization-based defense technique Tensor Network Purification(TNP) method, building upon tensor-based defense strategies. The approach significantly enhances the adversarial robustness of models.

**Strengths:**

This work presents a novel tensor network-based adversarial purification method, described in substantial detail, which significantly improves the computational efficiency of traditional TN approaches in AP tasks. The experimental analysis is comprehensive

**Weaknesses:**

1. The defense methods compared in Tables 1–4 vary significantly and lack consistency, which hinders the ability to draw unified conclusions. In particular, it is difficult to assess the specific impact of PTR on SA performance based on the presented comparisons.
2. In Section 4.1, it is mentioned that 512 images were randomly selected for testing. Could author clarify the rationale behind this specific sample size? Additionally, it would be helpful to provide further details regarding the selection process and whether all experiments were conducted exclusively on this subset. The limited sample size raises concerns about potential randomness in the evaluation.
3. Based on the experimental results, PTR appears to suffer from significant overfitting to the distribution of adversarial examples. This is evidenced by a notable improvement in both quantitative and qualitative performance after reconstruction, coupled with a clear decline in the SA metric.
4. Since the proposed defense method operates through a coarse-to-fine iterative process, an analysis of its computational efficiency is necessary.

**Questions:**

See weakness.

---

> ### Author Response · Authors · 2025-11-19
> **Responses to Reviewer tpP1 (1/2)**
>
> We sincerely appreciate Reviewer tpP1 for your time and constructive feedback. We are pleased that the Reviewer recognizes the **novelty**, **computational efficiency**, and **experimental comprehensiveness** of our work. The following are our responses to the comments and questions raised.
>
> Firstly, we would like to clarify one potential misunderstanding: In Weaknesses, the term ''PTR'' appears, although no such component exists in our paper. We believe this may have been a typo, likely referring to our proposed method (TNP). For clarity, in the following responses, we interpret ''PTR'' as ''TNP''.
>
> > **Weakness 1:** The defense methods compared in Tables 1–4 vary significantly and lack consistency. It is difficult to assess the specific impact of TNP on SA.
>
> **Response:** We would like to clarify that the results presented in Tables 1–4 are taken from the state-of-the-art (SOTA) methods in RobustBench ranking, which involve different norm threats and datasets. The compared defense methods differ because these SOTA defenses are originally designed and reported under different threats and datasets, which naturally results in different rankings, with different sets of defense methods appearing across the various settings in Tables 1–4.
>
> Regarding SA, our method matches or slightly exceeds state-of-the-art baselines on CIFAR-10. On CIFAR-100 and ImageNet, it remains comparable to some baselines, although it is moderately below the best-performing ones. Importantly, for all baselines with higher SA, our method consistently achieves higher RA. Considering that TNP requires no training, this level of performance represents a reasonable balance between standard accuracy and robustness.
>
> > **Weakness 2:** Could the authors clarify the rationale for using a randomly selected 512-image subset? The limited sample size raises concerns about potential randomness. More details on the selection process and whether all experiments used this subset would be helpful.
>
> **Response:** This protocol follows Nie et al. (ICML 2022), who demonstrated through extensive experiments that evaluating on a fixed 512-image subset yields results nearly identical to those obtained on the full test set, making it an efficient and reliable choice. This setting has since become common practice and is widely used for evaluating defense methods, as in [1–5]. To further alleviate the Reviewers’ concern, we repeat the experiments five additional times on different subsets, as shown below. The results demonstrate that evaluations on the 512-image subsets are stable, which further supports the above conclusion.
>
> |||||||||
> |:-:|:-:|:-:|:-:|:-:|:-:|:-:|:-:|
> | | Org.   | 1     | 2     | 3     | 4     | 5     |  Mean $\pm$ Std |
> | SA |  82.23 | 82.81 | 83.20 | 82.42 | 83.01 | 82.03 | 82.62 $\pm$ 0.46 |
> | RA |  55.27 | 54.88 | 55.66 | 54.68 | 55.85 | 56.05  | 55.40 $\pm$ 0.55 |
> |||||||||
>
> In the manuscript, we randomly selected 512 images at the beginning of our experiments and fixed this subset throughout the entire evaluation. All reported results were obtained using the same fixed set of images.
>
> [1] Chen et al. Robust Classification via a Single Diffusion Mode. ICML 2024.
> [2] Lin et al. Adversarial Training on Purification (AToP): Advancing Both Robustness and Generalization. ICLR 2024.
> [3] Zollicoffer et al. Lorid: Low-rank iterative diffusion for adversarial purification. AAAI 2025.
> [4] Li et al. Adversarial Diffusion Bridge Model for Reliable Adversarial Purification. ICLR 2025.
> [5] Wesego and Rooshenas. Adversary Aware Optimization for Robust Defense. NeurIPS 2025.

---

> ### Author Response · Authors · 2025-11-19
> **Responses to Reviewer tpP1 (2/2)**
>
> > **Weakness 3:**
> Based on the experimental results, TNP appears to suffer from significant overfitting to the distribution of adversarial examples. This is evidenced by a notable improvement in both quantitative and qualitative performance after reconstruction, coupled with a clear decline in the SA metric.
>
> **Response:** We respectfully point out that TNP does not suffer from overfitting.
> - As shown in Figure 4b, the reconstructed adversarial examples after applying TNP exhibit a significant reduction in adversarial perturbations. In contrast, the classical TN optimized with the traditional loss function shows overfitting, where the perturbations are restored, as illustrated in the second column of Figure 4b.
> - This point can be further supported by the quantitative results in the ''Adversarial example'' column of Table 7. TNP does not achieve notable improvements in reconstructing adversarial examples, and its reconstruction performance is weaker than that of other methods, but it achieves the highest robust accuracy.
> - Moreover, the drop in SA is not due to overfitting. In adversarial defense, purification methods inevitably alter some information in the input image to remove perturbations. This process often results in a reduction in SA, which is a commonly observed phenomenon in adversarial robustness.
>
> Taken together, the quantitative and qualitative results indicate that TNP does not suffer from overfitting.
>
> > **Weakness 4:** Since the proposed defense method operates through a coarse-to-fine iterative process, an analysis of its computational efficiency is necessary.
>
> **Response:** We appreciate the Reviewer's attention to computational efficiency. For the coarse-to-fine iterative process, the computational complexity is $\mathcal{O}(\log(D)\times r^3 \times 2^D)$ for each iteration, where $r$ is TN-rank and $2^D$ is the size of images, which is efficient. In the original manuscript, we also discussed the computational cost and provided detailed results in Table 10, along with further analysis in Appendix E.2. The results indicate that the proposed method is competitive compared with other AP methods.

---

### Author Response · Authors · 2025-11-27

Dear Reviewers,

As the discussion period is now less than a week from concluding, we hope to receive your feedback and that your concerns are adequately addressed. If you have any remaining questions or would like further clarification, we will do our best to address them before the period concludes.

---

### Meta-Review · Area_Chair_HNda · 2026-01-07

**Summary:**

There are four reviewers for this paper, with the initial rating of 8,6,4,,4.

Reviewer **tpP1** recognizes the novelty and comprehension of the experiments. The main concerns include the lack of a specific impact of PTR on SA performance based on the presented comparisons and the computational efficiency issue.

Reviewer **2dCC**’s main concerns include the evaluation against adaptive attacks, the proposed method's efficiency, and its experimental performance. I think those concerns are not well solved by the rebuttal.

Reviewer **xrYF**’s main concerns include the writing of the paper and the problem statement.

Reviewer **zp4E** gave a quite positive opinion on this paper, and explicitly indicates that most of his/her concerns have been addressed.

**Reviewer Concerns:**

After reading the comments and responses, I think some concerns are addressed well, but Reviewer **2dCC**’s concerns are not addressed well, including the efficiency and the performance.

**Reviewer Scores:**

I think Reviewer **2dCC** will keep the negative rating.

---

### Decision · Program_Chairs · 2026-01-26

Reject